# Tracking the Median of Gradients with a Stochastic Proximal Point Method

**Fabian Schaipp**  *fabian.schaipp@inria.fr*
*Technical University of Munich and Inria Paris, ENS, PSL Research University*

**Guillaume Garrigos**  *garrigos@lpsm.paris*
*Université Paris Cité and Sorbonne Université, CNRS*
*Laboratoire de Probabilités, Statistique et Modélisation, F-75013 Paris, France*

**Umut Şimşekli**  *umut.simsekli@inria.fr*
*Inria, CNRS, ENS, PSL Research University*
*Paris*

**Robert M. Gower**  *rgower@flatironinstitute.org*
*CCM, Flatiron Institute, Simons Foundation*
*New York City*

**Reviewed on OpenReview:** *https://openreview.net/forum?id=WMxLEgYGxu*

## Abstract

There are several applications of stochastic optimization where one can benefit from a robust estimate of the gradient. For example, domains such as distributed learning with corrupted nodes, the presence of large outliers in the training data, learning under privacy constraints, or even heavy-tailed noise due to the dynamics of the algorithm itself. Here we study `SGD` with robust gradient estimators based on estimating the median.

We first derive iterative methods based on the stochastic proximal point method for computing the median gradient and generalizations thereof. Then we propose an algorithm estimating the median gradient across *iterations*, and find that several well known methods are particular cases of this framework. For instance, we observe that different forms of clipping allow to compute online estimators of the *median* of gradients, in contrast to (heavy-ball) momentum, which corresponds to an online estimator of the *mean*. Finally, we provide a theoretical framework for an algorithm computing the median gradient across *samples*, and show that the resulting method can converge even under heavy-tailed, state-dependent noise.

## 1 Introduction

Many problems in machine learning can be represented by an optimization problem

$$\min_{\boldsymbol{w} \in \mathbb{R}^d} \ell(\boldsymbol{w}),$$

where $\boldsymbol{w}$ is a vector of parameters and $\ell : \mathbb{R}^d \to \mathbb{R}$ is a loss function. To tackle this problem, gradient-based optimization algorithms have been the de-facto choice in many application domains (Nemirovsky & Yudin, 1983; Bottou, 2010; Bottou et al., 2018), where we are only given access to a *noisy oracle* of the gradient

$$\boldsymbol{g}_t = \nabla \ell(\boldsymbol{w}_t) + \boldsymbol{\xi}_t(\boldsymbol{w}_t). \tag{1}$$

Here $t$ denotes the iterations and $\boldsymbol{\xi}_t(\boldsymbol{w}_t) \in \mathbb{R}^d$ is a noise vector that is sampled from a distribution that depends on the parameter $\boldsymbol{w}_t$. With this noise oracle, stochastic gradient descent (SGD) with learning rate $\eta_t$ is given by

$$\boldsymbol{w}_{t+1} = \boldsymbol{w}_t - \eta_t \boldsymbol{g}_t. \tag{SGD}$$

While SGD has proven useful in numerous applications, its performance heavily relies on the 'quality' of the noisy oracles $\boldsymbol{g}_t$, and can tragically degrade when the noise $\boldsymbol{\xi}_t$ has significant outliers, exhibits heavy tails (that is, $\|\boldsymbol{\xi}_t\|$ is very large with non-negligible probability), or has been corrupted in an adversarial way. We provide two examples.

(i) **Heavy-tailed gradients.** Recent studies have provided theoretical and empirical evidence that heavy tails can naturally arise in stochastic optimization. From a theoretical perspective Gurbuzbalaban et al. (2021); Hodgkinson & Mahoney (2021); Lehman et al. (2023) showed that the parameter-dependent nature of $\boldsymbol{\xi}_t$ can result in heavy tails, even in very basic settings such as linear regression with Gaussian data. On the other hand, it has been empirically observed that the gradients for training transformer architectures on language tasks are more heavy-tailed compared to, for example, convolutional models for image data (Zhang et al., 2020b; Kunstner et al., 2023). While the presence of heavy-tails in gradients might be beneficial in certain settings (Simsekli et al., 2020), from an optimization perspective, it mostly introduces non-trivial challenges, which might even make the algorithm diverge unless additional care is taken (Zhang et al., 2020b; Gorbunov et al., 2020).

(ii) **Corrupted nodes.** A well-known problem in distributed learning is when some computation nodes are malicious and can communicate adversarial updates, which results in inaccurate gradient oracles that can significantly misguide the optimization procedure. Similar to the heavy-tailed setting, depending on how malicious the nodes are, SGD can be impractical unless certain modifications are made (Mhamdi et al., 2018). We refer to Karimireddy et al. (2021) and references therein for an overview of more robust aggregation techniques.

In order to make the optimization algorithms more robust to such noisy oracles, many techniques have been developed for different domains under various conditions. In the context of heavy-tailed gradients, clipping is often employed (Zhang et al., 2020a; Puchkin et al., 2023; Koloskova et al., 2023). In distributed learning robust aggregation rules are needed to protect against corrupted or malicious nodes (Data et al., 2018; Karimireddy et al., 2021; Khirirat et al., 2023). This line of research is also closely related to error feedback in federated learning (Seide et al., 2014; Karimireddy et al., 2019; Richtarik et al., 2021) and learning under differential privacy constraints (Khirirat et al., 2023).

**Robustness of median.** While attracting increasing attention in stochastic optimization thanks to modern machine learning applications, taming the impact of strong noise has already been considered in various other domains. Indeed, estimation under heavy-tails and corruptions has been a long-standing topic in robust statistics (Huber, 1981). Being a vast research field that has produced numerous algorithms designed for different tasks, one of the recurring themes in robust statistics is the use of the *sample median* and its variations as opposed to the *sample mean* (Minsker, 2015), whenever an aggregation of random variables is needed. For instance, in the presence of heavy tails, the sample median has been shown to be a significantly more robust estimator of the true mean, whereas the sample mean can be vulnerable to strong noise (Lugosi & Mendelson, 2019).

**Sample median SGD.** Inspired by tools from robust statistics, some notions of median have been utilized in stochastic optimization as well (Yin et al., 2018; Alistarh et al., 2018; Acharya et al., 2022). One such approach is to compute the median over a finite sample of gradients. While this approach has paved the way for powerful optimization algorithms in terms of robustness, it often introduces a significant computational burden, since a multivariate median needs to be computed at every iteration (Weiszfeld & Plastria, 2009; Vardi & Zhang, 2000).

**Motivation.** Clipping-based and error feedback approaches on the one hand, and median-based approaches on the other hand are motivated through different mathematical frameworks and appear to use different

algorithmic tools. In this study, our main goal is to bridge the gap between these two seemingly different branches and to bring a unified theoretical perspective that can shed further light on both directions. The main tool to build this bridge between stochastic optimization and robust statistics will be the Stochastic Proximal Point method (SPP) (Asi & Duchi, 2019; Davis & Drusvyatskiy, 2019; Toulis et al., 2020) for solving a later-specified gradient estimation problem.

**Contributions.** We first show how heavy-ball momentum (Polyak, 1964) can be seen as online estimator of the mean gradient using SPP. Here we use the term *online* in the sense that the method is given a different loss function at each step.

Encouraged by this result, we then focus on online estimation of the *median* gradient, using the same technique. Doing so, we recover several known clipping-based optimization algorithms from distributed learning as special cases, such as Clip21 (Khirirat et al., 2023), and a variant of *centered clipping* (Karimireddy et al., 2021). We also shown connections to sign-based gradient methods (Riedmiller & Braun, 1993).

Our proposed framework allows to unify the different motivations for previously developed techniques, and illustrates that clipping-based optimization algorithms essentially run a loose estimation of the median across iterations. In particular, the observation that *centered clipping* and Clip21 are iterative estimators of a geometric median appears to be new, and might have consequences for its application in distributed learning.

We theoretically analyze these online gradient estimation methods in a simplified setting: when allowing multiple gradient samples per iteration, we show that SGD with (approximately computed) median gradients indeed yields a very powerful algorithm, which converges with $1/\sqrt{T}$ rate ($T$ being the number of iterations) even when the gradient oracles are heavy-tailed with diverging second-order moments. We further illustrate that, in addition to having infinite variance, the noise vectors $\boldsymbol{\xi}_t(\boldsymbol{w})$ can even have a multiplicative dependence on $\boldsymbol{w}$, a scenario which is highly challenging and cannot be directly covered by existing theoretical results, see e.g., Wang et al. (2021); Zhang et al. (2020b); Puchkin et al. (2023).

We finally illustrate our theory on synthetic least-squares experiments where we compare the effectiveness of sample median, sample mean, and several online median estimators. Our results underline that for heavy-tailed noise, using the sample median is highly effective in contrast to the sample mean which is unstable and often does not converge. Our experiments also show that our online median estimates are robust, and require only a single sample per iteration, making it a less expensive alternative to the sample median. We further compare different clipping techniques for training transformer architectures on language modeling tasks, and show that they can improve upon the performance of SGD with momentum, however the gap is relatively small.

**Notation.** Throughout the paper, $\|\cdot\|_p$ denotes the $\ell_p$-norm, given by $\|\boldsymbol{z}\|_p := \left(\sum_{i=1}^d |\boldsymbol{z}|_i^p\right)^{\frac{1}{p}}$ when $p \in [1,\infty)$, and $\|\boldsymbol{z}\|_\infty = \max_{i=1,\ldots,d} |\boldsymbol{z}_i|$ when $p = \infty$. When $p = 2$ we recover the usual Euclidean $\ell_2$-norm $\|\cdot\|_2$, which we will often simply denote by $\|\cdot\|$. For a matrix $\boldsymbol{A}$, we denote its Frobenius norm by $\|\boldsymbol{A}\|_F := \sqrt{\sum_{i,j} \boldsymbol{A}_{ij}^2}$. For $n \in \mathbb{N}_+$, we denote the set $[n] := \{1, \ldots, n\}$.

## 2 Preliminaries

### 2.1 Heavy-tailed Random Variables

A random variable $X$ is said to be *heavy-tailed* if the tails of its distribution decay slower than the tails of an exponential distribution. One important example is the family of symmetric $\alpha$-stable distributions $\mathcal{P}_{\alpha,\sigma}$ (Nolan, 2020, Def. 1.4), parameterized by a stability index $\alpha \in (0,2]$ and a scale $\sigma > 0$. When $\alpha = 1$, this is the Cauchy distribution; when $\alpha = 2$, it reduces to a Gaussian. The parameter $\sigma$ controls the spread of the distribution. When $\sigma = 1$, we call the distribution *standard* and denote it by $\mathcal{P}_\alpha := \mathcal{P}_{\alpha,1}$. The parameter

$\alpha$ on the other hand controls the heaviness of the tails: for $\alpha < 2$ the random variable $X \sim \mathcal{P}_\alpha$ has infinite variance, whereas for $\alpha \leq 1$ even the mean of $X$ is infinite. However, for any $\alpha$, the median is equal to 0.

## 2.2 Mean Estimation and (Sample) Median

Estimating the mean of a distribution, when given a finite number of samples, is one of the core problems studied in statistics. If the distribution in question has heavy-tails, or has significant outliers, the sample mean turns out to be a poor estimator. Lugosi & Mendelson (2019) give a survey of alternative and more robust techniques: one such alternative is the median, for which we will introduce basic definitions next.

Here, we denote by $z$ a real-value random value, and the boldface $\boldsymbol{z} \in \mathbb{R}^d$ a random vector. We always denote the expectation with respect to the distribution of $z$ (or $\boldsymbol{z}$ respectively) by $\mathbb{E}$.

**Multivariate median.** The one-dimensional median is defined by

$$\texttt{median}(z) \in \operatorname*{argmin}_{m \in \mathbb{R}} \mathbb{E}\left[|m - z|\right]. \tag{2}$$

In contrast to the mean, the median is always defined (though it may not be unique) (Cramér, 2016). Since we are interested in approximating the median gradient, we need a notion of median that generalizes to random vectors $\boldsymbol{z} \in \mathbb{R}^d$. For this, we can define the $\ell_p$-median:

$$\texttt{median}_{\ell_p}(\boldsymbol{z}) \in \operatorname*{argmin}_{\boldsymbol{m} \in \mathbb{R}^d} \mathbb{E}\left[\|\boldsymbol{m} - \boldsymbol{z}\|_p\right], \tag{3}$$

where $\|\boldsymbol{m}\|_p = \left(\sum_{j=1}^d \boldsymbol{m}_j^p\right)^{\frac{1}{p}}$ is the $\ell_p$ norm. This recovers several existing notions of a multivariate median: for $p = 1$ we obtain a *componentwise median* (or $\ell_1$-median) and for $p = 2$ we obtain the *geometric median* (or $\ell_2$-median) (Fréchet, 1948; Weiszfeld & Plastria, 2009; Minsker, 2015; Cohen et al., 2016). For $d = 1$ and every $p \in [1, \infty)$, the $\ell_p$-median coincides with the one-dimensional median in (2).

**Sample median.** Given $n$ realizations $\boldsymbol{z}^{(1)}, \ldots, \boldsymbol{z}^{(n)}$ of $\boldsymbol{z}$, the sample median is defined as

$$\texttt{median}_{\ell_p, i \in [n]}(\boldsymbol{z}^{(i)}) \in \operatorname*{argmin}_{\boldsymbol{m} \in \mathbb{R}^d} \frac{1}{n} \sum_{i=1}^n \|\boldsymbol{m} - \boldsymbol{z}^{(i)}\|_p. \tag{4}$$

Clearly (4) is a special case of (3), by setting the distribution of $\boldsymbol{z}$ as the empirical measure $\frac{1}{n}\sum_{i=1}^n \delta_{\boldsymbol{z}_i}$.

It is generally accepted that the sample median is a more robust estimator than the sample mean (Huber, 1981; Hampel et al., 2005). This is nicely illustrated by the notion of a breakdown point (Donoho & Huber, 1983; Lopuhaa & Rousseeuw, 1991), which is the smallest fraction of a sample that, if arbitrarily corrupted, can arbitrarily change the value of the estimator. The breakdown point of the sample median is roughly $\frac{1}{2}$, meaning that at least half of the samples need to be outliers for the median to diverge. The sample mean however can be arbitrarily corrupted by a single outlier, thus its breakdown point is $\frac{1}{n}$.

If the samples $\boldsymbol{z}^{(i)}$ are obtained by taking block-wise means over observations, (4) is called the median-of-means estimator (Nemirovsky & Yudin, 1983; Lugosi & Mendelson, 2019), which is known to be a robust mean estimator (Lugosi & Mendelson, 2019).

**Relation to M-estimation.** More generally, for a function $\mathcal{D} : \mathbb{R}^d \to \mathbb{R}$ we will consider the problem

$$\operatorname*{argmin}_{\boldsymbol{m} \in \mathbb{R}^d} \mathbb{E}\left[\mathcal{D}(\boldsymbol{m} - \boldsymbol{z})\right]. \tag{5}$$

We assume that $\mathcal{D}$ is a closed, proper, convex function and that (5) admits a solution (see Appendix B for a discussion). For $\mathcal{D} = \frac{1}{2}\|\cdot\|_2^2$, the solution to (5) is the mean $\mathbb{E}[\boldsymbol{z}]$ (if it exists), and for $\mathcal{D} = \|\cdot\|_2$ the solution

is the geometric median as defined in (3). Problem (5) can be seen as a special case of *M-estimators* (Huber, 1981, Sec. 3.2).

The choice of $\mathcal{D}$ determines properties of the associated estimator: if the function $\mathcal{D}$ heavily penalizes large values, like the quadratic $\mathcal{D} = \frac{1}{2}\|\cdot\|_2^2$, then the solution to (5) will be sensitive to outliers. In contrast $\mathcal{D} = \|\cdot\|_2$ and $\mathcal{D} = \|\cdot\|_1$ grow only linearly, and thus (5) will be less sensitive to outliers. We return to problem (5) later in Section 3.2.

## 3 Robust Gradient Estimation with Stochastic Proximal Point

### 3.1 Warmup: Momentum as Online Mean Gradient Estimation

When dealing with noisy gradient oracles, a standard technique is to apply (heavy-ball) momentum (Polyak, 1964). For momentum coefficient $\beta \in [0, 1)$, it is given by

$$\begin{aligned} \boldsymbol{m}_{t+1} &= \beta\boldsymbol{m}_t + (1-\beta)\boldsymbol{g}_t, \\ \boldsymbol{w}_{t+1} &= \boldsymbol{w}_t - \eta_t\boldsymbol{m}_{t+1}. \end{aligned} \tag{SGD-M}$$

Momentum has been empirically shown to improve the performance of `SGD` in machine learning (Sutskever et al., 2013), and can be seen as a variance reduction technique (Gower et al., 2020).

It is easy to see that momentum performs an exponentially weighted average over past gradients. In this section, we will derive a new perspective on `SGD-M`, namely being an *online estimator of the mean gradient* at a given point. To see this, we recall that for a random variable $\boldsymbol{z} \in \mathbb{R}^d$, its mean (if it exists) is the solution to (5) with $\mathcal{D} = \frac{1}{2}\|\cdot\|^2$. The random variable of interest in this context are the noisy gradients $\boldsymbol{g}_t$. We could in principle solve problem (5) with iterative stochastic methods. For this purpose, we consider the stochastic proximal point (`SPP`) method (Asi & Duchi, 2019; Davis & Drusvyatskiy, 2019). We give a detailed introduction to `SPP` in the next section.

For the purpose of this section, it is sufficient to know that the iterates $(\boldsymbol{m}_t)$ of `SPP` with step-size $\tau > 0$ applied to (5) are given by (details in Appendix B.2)

$$\boldsymbol{m}_{t+1} = \operatorname*{argmin}_{\boldsymbol{m}} \mathcal{D}(\boldsymbol{m} - \boldsymbol{g}_t) + \frac{1}{2\tau}\|\boldsymbol{m} - \boldsymbol{m}_t\|^2 = \boldsymbol{g}_t + \operatorname{prox}_{\tau\mathcal{D}}(\boldsymbol{m}_t - \boldsymbol{g}_t), \tag{6}$$

where we denote by $\operatorname{prox}_{\mathcal{D}}$ the proximal operator of $\mathcal{D}$,

$$\operatorname{prox}_{\mathcal{D}}(\boldsymbol{x}) := \operatorname*{argmin}_{\boldsymbol{y}\in\mathbb{R}^d} \mathcal{D}(\boldsymbol{y}) + \frac{1}{2}\|\boldsymbol{y} - \boldsymbol{x}\|^2.$$

For $\mathcal{D} = \frac{1}{2}\|\cdot\|^2$, it is easy to compute $\operatorname{prox}_{\tau\mathcal{D}}(\boldsymbol{x}) = \frac{1}{1+\tau}\boldsymbol{x}$. Thus, one iteration of `SPP` (6) is equal to

$$\boldsymbol{m}_{t+1} = \boldsymbol{g}_t + \operatorname{prox}_{\tau\mathcal{D}}(\boldsymbol{m}_t - \boldsymbol{g}_t) = \boldsymbol{g}_t + \frac{1}{1+\tau}(\boldsymbol{m}_t - \boldsymbol{g}_t) = (1 - \frac{\tau}{1+\tau})\boldsymbol{m}_t + \frac{\tau}{1+\tau}\boldsymbol{g}_t. \tag{7}$$

This is exactly the momentum step in (`SGD-M`) with $\beta = 1 - \frac{\tau}{1+\tau}$. However, in `SGD-M` the parameters $\boldsymbol{w}_t$ are updated each step as well, and thereby the distribution of the gradients $\boldsymbol{g}_t$ changes each step. In conclusion, the momentum operation can be understood as an *online* estimation of the mean gradient.

We end this section by justifying our choice to specifically consider `SPP` for solving (5), instead of `SGD` or other stochastic methods. If we were to apply `SGD` with step size to problem (5), we would obtain

$$\boldsymbol{m}_{t+1} = (1 - \tau)\boldsymbol{m}_t + \tau\boldsymbol{g}_t.$$

This also resembles momentum, but note that the momentum coefficient $1 - \tau$ could be negative if $\tau > 1$. For `SPP` on the other hand, the momentum coefficient is correctly parametrized since $1 - \frac{\tau}{1+\tau}$ is contained in $[0, 1)$ for every $\tau > 0$.

In conclusion, we can understand heavy-ball momentum (`SGD-M`) as applying `SPP` to the *mean* estimation problem in an *online* fashion. This opens the question what is the `SPP` update for estimators that are more robust to heavy-tailed noise, such as the median (or more generally for other choices for $\mathcal{D}$ that are of interest). We answer this in the next section.

### 3.2 Clipping as Online Median Gradient Estimation

We now focus on the following gradient estimation problem: let $\mathcal{G}$ denote the (fixed) distribution over noisy gradients $\boldsymbol{g}$. For $\mathcal{D} : \mathbb{R}^d \to \mathbb{R}_{\geq 0}$ being a closed, proper, convex function, we consider the problem of approximating

$$\operatorname*{argmin}_{\boldsymbol{m}\in\mathbb{R}^d} \mathbb{E}_{\boldsymbol{g}\sim\mathcal{G}} \left[ \mathcal{D}(\boldsymbol{m} - \boldsymbol{g}) \right]. \tag{8}$$

We consider stochastic iterative methods for solving (8), with a special focus on the stochastic proximal point method (`SPP`). We recall the `SPP` update being

$$\begin{aligned} \boldsymbol{m}_{t+1} &= \operatorname*{argmin}_{\boldsymbol{m}} \mathcal{D}(\boldsymbol{m} - \boldsymbol{g}_t) + \frac{1}{2\tau}\|\boldsymbol{m} - \boldsymbol{m}_t\|^2 \\ &= \boldsymbol{g}_t + \operatorname{prox}_{\tau\mathcal{D}}(\boldsymbol{m}_t - \boldsymbol{g}_t). \end{aligned} \tag{9}$$

The `SPP` update can be seen as an implicit version of `SGD`, which is hard to compute in general. In cases where `SPP` has a closed form update, it is often preferred to `SGD` since is easier to tune (Asi & Duchi, 2019; Milzarek et al., 2024). Fortunately, as we show next, the `SPP` method enjoys closed-form updates for all choices of $\mathcal{D}$ which are of interest in our context, in particular for $\mathcal{D} \in \{\|\cdot\|_1, \|\cdot\|_2, \frac{1}{2}\|\cdot\|^2\}$. All the details are deferred to Appendix B.3.

**Vectorwise clipping.** If we choose $\mathcal{D} = \|\cdot\|_2$, the `SPP` update applies clipping to the difference between sampled gradient $\boldsymbol{g}_t$ and previous estimate $\boldsymbol{m}_t$.

> **Corollary 3.1.** For $\mathcal{D} = \|\cdot\|_2$ update (9) is given by
>
> $$\boldsymbol{m}_{t+1} = \boldsymbol{m}_t + \operatorname{clip}_{\tau,2}(\boldsymbol{g}_t - \boldsymbol{m}_t), \tag{10}$$
>
> where $\operatorname{clip}_{\tau,2}(\boldsymbol{v}) := \frac{\tau}{\max\{\tau,\|\boldsymbol{v}\|_2\}}\boldsymbol{v}$.

We can rewrite update (10) as $\boldsymbol{m}_{t+1} = \beta_t\boldsymbol{m}_t + (1 - \beta_t)\boldsymbol{g}_t$ with $\beta_t := 1 - \frac{\tau}{\max\{\tau,\|\boldsymbol{g}_t-\boldsymbol{m}_t\|_2\}}$. In other words, online estimation of the geometric median ($\mathcal{D} = \|\cdot\|_2$) with `SPP` is equal to a cautious version of momentum, as samples which are too far from $\boldsymbol{m}_t$ will be down weighted. This allows the iterates to be more robust to outliers.

**Componentwise clipping.** If we choose $\mathcal{D} = \|\cdot\|_1$, `SPP` performs componentwise clipping instead. Here as well, clipping protects against large individual entries by shrinking them independently.

> **Corollary 3.2.** For $\mathcal{D} = \|\cdot\|_1$ update (9) is given by
>
> $$\boldsymbol{m}_{t+1} = \boldsymbol{m}_t + \operatorname{clip}_{\tau,1}(\boldsymbol{g}_t - \boldsymbol{m}_t), \tag{11}$$
>
> where $\operatorname{clip}_{\tau,1}(\boldsymbol{v}) := (\min\{\max\{\boldsymbol{v}_i, -\tau\}, \tau\})_{i=1}^d$.

**Huber function.** For update (10), it is possible that $\boldsymbol{m}_{t+1} = \boldsymbol{g}_t$, thus no momentum. To allow for a soft transition, we can choose $\mathcal{D}$ to be the Huber function. For $\mu > 0$, it is given by

$$H_\mu : \mathbb{R}^d \to \mathbb{R}, \quad H_\mu(\boldsymbol{z}) = \begin{cases} \frac{1}{2}\|\boldsymbol{z}\|^2 & \|\boldsymbol{z}\| \leq \mu, \\ \mu\|\boldsymbol{z}\| - \frac{\mu^2}{2} & \text{else.} \end{cases}$$

The Huber function operates like the squared loss for arguments with small norm, and like the $\ell_2$-norm for arguments with large norms (potentially outliers).

**Corollary 3.3.** For $\mathcal{D} = H_\mu$, update (9) is given by

$$\boldsymbol{m}_{t+1} = \boldsymbol{g}_t + \beta_t(\boldsymbol{m}_t - \boldsymbol{g}_t) = \beta_t \boldsymbol{m}_t + (1 - \beta_t)\boldsymbol{g}_t, \tag{12}$$

where $\beta_t := 1 - \frac{\mu\tau}{\max\{\|\boldsymbol{m}_t - \boldsymbol{g}_t\|, \ \mu(1+\tau)\}}$.

In update (12) the coefficient $\beta_t$ is always positive. Thus, $\mathcal{D} = H_\mu$ is a tradeoff between momentum with fixed coefficient ($\mathcal{D} = \frac{1}{2}\|\cdot\|^2$) and clipping ($\mathcal{D} = \|\cdot\|_2$). However, it comes at the cost of choosing/tuning an additional hyperparameter $\mu$. In Appendix B.4, we additionally compare the updates when using stochastic subgradient descent instead of SPP for solving (8), showing that sign-SGD is related to $\ell_1$-median estimation.

## 4 Gradient Estimator in the Wild

Our final objective is to use robust gradient estimators within a SGD-type method. Because at each iteration the weights $\boldsymbol{w}_t$ are updated, the distribution of the gradients (denoted by $\mathcal{G}$ in the previous section) also changes at each iteration of SGD. Our strategy is to interweave updates in the weights $\boldsymbol{w}_t$ with SPP updates in the gradient estimators $\boldsymbol{m}_t$. That is, we sample *one* stochastic gradient $\boldsymbol{g}_t$ per iteration and set

$$
\begin{aligned}
\boldsymbol{m}_{t+1} &= \boldsymbol{g}_t + \mathrm{prox}_{\tau\mathcal{D}}(\boldsymbol{m}_t - \boldsymbol{g}_t), \\
\boldsymbol{w}_{t+1} &= \boldsymbol{w}_t - \eta_t \boldsymbol{m}_{t+1}.
\end{aligned}
\tag{13}
$$

With the results from Sections 3.1 and 3.2, we immediately obtain the following special cases:

(i)  if $\mathcal{D} = \frac{1}{2}\|\cdot\|^2$ : $\begin{cases} \boldsymbol{m}_{t+1} = (1 - \frac{\tau}{1+\tau})\boldsymbol{m}_t + \frac{\tau}{1+\tau}\boldsymbol{g}_t \\ \boldsymbol{w}_{t+1} = \boldsymbol{w}_t - \eta_t \boldsymbol{m}_{t+1}, \end{cases}$

(ii)  if $\mathcal{D} = \|\cdot\|_p, \ p \in \{1, 2\}$ : $\begin{cases} \boldsymbol{m}_{t+1} = \boldsymbol{m}_t + \mathrm{clip}_{p,\tau}(\boldsymbol{g}_t - \boldsymbol{m}_t) \\ \boldsymbol{w}_{t+1} = \boldsymbol{w}_t - \eta_t \boldsymbol{m}_{t+1}. \end{cases}$

Equation (i) is SGD with momentum where the SPP step size $\tau$ defines the momentum coefficient. On the other hand, (ii) corresponds to clipping, and here the SPP step size determines the clipping threshold. In other words, SGD with momentum is to online mean estimation what clipping is to online median estimation.

A slightly different approach would be to restart $\boldsymbol{m}_t = \boldsymbol{0}$ in every iteration (*cold start*). In this case, (i) reduces to SGD with no momentum, while in (ii) the clipping operator is applied to $\boldsymbol{g}_t$, instead of $\boldsymbol{g}_t - \boldsymbol{m}_t$, which is also a common technique in practice (Zhang et al., 2020a). We discuss this relation below in more detail.

**Connection to Error Feedback.** Methods such as (13) are also used in the distributed learning setting to improve communication and protect against corrupted nodes (Karimireddy et al., 2021). These methods are often called Error Feedback (EF) methods (Seide et al., 2014; Richtarik et al., 2021) and generally involve an update of the form $\boldsymbol{m}_{t+1} = \boldsymbol{m}_t + \mathcal{C}_t(\boldsymbol{g}_t - \boldsymbol{m}_t)$ where $\mathcal{C}_t : \mathbb{R}^d \mapsto \mathbb{R}^d$ is a compression operator. Projections onto balls can be seen as a compression, since one can encode the resulting vector in fewer bits (Safaryan et al., 2021). In particular, centered clipping by Karimireddy et al. (2021), and the Clip21 method (Khirirat et al., 2023) are using variations of update (10) in the distributed setting, and are thus 'secretly estimating' the geometric median of the gradients returned across all nodes.

**SGD with standard clipping.** Even in the non-distributed setting, a standard technique to avoid training instabilities is to directly clip the gradient $\boldsymbol{g}_t$ (Zhang et al., 2020a). For momentum coefficient $\beta \in [0, 1)$ and

$c > 0$, let

$$
\begin{aligned}
\boldsymbol{m}_{t+1} &= \beta \boldsymbol{m}_t + (1 - \beta) \min\{1, \tfrac{c}{\|\boldsymbol{g}_t\|}\} \boldsymbol{g}_t, \\
\boldsymbol{w}_{t+1} &= \boldsymbol{w}_t - \eta_t \boldsymbol{m}_{t+1}.
\end{aligned}
\tag{14}
$$

We will refer to (14) as `clipped-SGD`. The method has been studied extensively, often for the case $\beta = 0$, see Zhang et al. (2020a); Gorbunov et al. (2020); Koloskova et al. (2023); Puchkin et al. (2023) and references therein.

As explained above, when $\beta = 0$, then (14) is the same as (10) with cold start (resetting $\boldsymbol{m}_t = \boldsymbol{0}$ in every iteration); but this does not allow for momentum, which is usually used in practice on top of clipping. However, we can derive `clipped-SGD` (14) also as online SPP for computing a specific robust mean estimator, known as the *Catoni-Giulini mean estimator* (Catoni & Giulini, 2018; Lugosi & Mendelson, 2019). For this, consider the problem

$$
\operatorname*{argmin}_{\boldsymbol{m} \in \mathbb{R}^d} \mathbb{E}_{\boldsymbol{g} \sim \mathcal{G}} \left[ \tfrac{1}{2} \| \boldsymbol{m} - \min\{1, \tfrac{c}{\|\boldsymbol{g}\|}\} \boldsymbol{g} \|^2 \right].
\tag{15}
$$

The solution to (15) is $\mathbb{E}_{\boldsymbol{g} \sim \mathcal{G}} \left[ \min\{1, \tfrac{c}{\|\boldsymbol{g}\|}\} \boldsymbol{g} \right]$, which is exactly the Catoni-Giulini mean estimator. Now, one step of SPP (9) applied to (15) gives (14) with $\beta = \frac{1}{1+\tau}$.

**Outline of remaining paper.** Before comparing these different online gradient estimation methods in experiments, we will discuss convergence guarantees: ideally, we would like to analyze the online method (13). However, when $\mathcal{D}$ is not strongly convex this analysis turns out to be difficult, mainly due to the fact that the distribution of $\boldsymbol{g}_t$ changes every step. Hence, the next section will analyze the online median estimator in a simplified setting, where we have access to multiple samples each iteration, and can (approximately) compute their sample median. We focus on settings where the noise in $\boldsymbol{g}_t$ is heavy-tailed, and prove that in such scenarios the sample median can still guarantee convergence (while the sample mean fails).

Finally, our experiments in Section 6 will show that the online versions of the sample median method (`VClip` and `CClip`) show superior performance as well in the heavy-tailed scenario, justifying the online estimation.

## 5 Theory for the Sample Median Gradient Method

In this section, we showcase that `SGD` with an (approximate) sample median is robust to heavy-tailed noise, whereas `SGD` using the sample mean can fail. Consider the following setup: at every iteration $t \geq 0$, we have access to $n$ noisy oracle gradients $\big(\boldsymbol{g}_t^{(1)}, \ldots, \boldsymbol{g}_t^{(n)}\big)$, where

$$
\boldsymbol{g}_t^{(i)} = \nabla \ell(\boldsymbol{w}_t) + \boldsymbol{\xi}_t^{(i)}(\boldsymbol{w}_t) \quad \text{for} \quad i = 1, \ldots, n.
$$

Here $\{\boldsymbol{\xi}_t^{(i)}(\boldsymbol{w}_t) \in \mathbb{R}^d\}_{i=1}^n$ is a set of i.i.d. noise vectors. Given these sampled gradients, the typical approach of SGD would be to use their sample mean (average) $\boldsymbol{g}_t = \frac{1}{n} \sum_{i=1}^n \boldsymbol{g}_t^{(i)}$ as update direction. However, Zhang et al. (2020b, Remark 1) show that this fails to converge for any step size under heavy-tailed noise.

Alternatively, to improve robustness, we analyze the update

$$
\boldsymbol{w}_{t+1} = \boldsymbol{w}_t - \eta(\boldsymbol{m}_t + \boldsymbol{e}_t),
\tag{16}
$$

where $\boldsymbol{m}_t = \texttt{median}_{\ell_p, i \in [n]}(\boldsymbol{g}_t^{(i)})$, and $\boldsymbol{e}_t \in \mathbb{R}^d$ represents a random error we may commit in computing the sample median. We refer to method (16) as *Sample Median Gradient Descent* (`SMGD`).

**Assumption 5.1.** There exists $\sigma_1, \sigma_2, \delta_1, \delta_2 \geq 0$ such that, for every $\boldsymbol{w} \in \mathbb{R}^d$, the sample median $\boldsymbol{m} = \texttt{median}_{\ell_p, i \in [n]}(\boldsymbol{g}^{(i)})$ verifies

$$
\begin{aligned}
\|\mathbb{E}[\boldsymbol{m}] - \nabla\ell(\boldsymbol{w})\|^2 &\leq \delta_1^2 + \delta_2 \|\nabla\ell(\boldsymbol{w})\|^2, \\
\mathbb{E}\left[\|\boldsymbol{m} - \mathbb{E}[\boldsymbol{m}]\|^2\right] &\leq \sigma_1^2 + \sigma_2 \|\nabla\ell(\boldsymbol{w})\|^2.
\end{aligned}
\tag{17}
$$

**Assumption 5.2.** There exists $\varepsilon > 0$ such that, for every $t \geq 0$, the error in computing the sample median has finite variance: $\mathbb{E}\left[\|\boldsymbol{e}_t\|^2 \mid \boldsymbol{g}_t^{(1)}, \ldots, \boldsymbol{g}_t^{(n)}\right] \leq \varepsilon^2$.

Based on these two assumptions, we establish the following complexity results for (16).

**Proposition 5.3.** Let $\ell : \mathbb{R}^d \to \mathbb{R}$ be $L$-smooth and let Assumption 5.1 hold with $\delta_2 \leq \frac{1}{8}$. Take $\eta \leq \frac{1}{8L(1+2\sigma_2+2\delta_2)}$ and consider $\boldsymbol{w}_t$ a sequence generated by (16) where $\boldsymbol{e}_t$ verifies Assumption 5.2. Then, for every $T \geq 1$ it holds

$$
\min_{0 \leq t \leq T-1} \mathbb{E}\left[\|\nabla\ell(\boldsymbol{w}_t)\|^2\right] = \mathcal{O}\left(\frac{1}{\eta T} + \eta + \delta_1^2 + \varepsilon^2\right).
$$

**Proposition 5.4.** Let $\ell : \mathbb{R}^d \to \mathbb{R}$ be convex and $G$-Lipschitz, with $\operatorname{argmin} \ell \neq \emptyset$. Let Assumptions 5.1 and 5.2 hold. Without loss of generality assume that $\delta_2 = \sigma_2 = 0$. Then, for $\eta = \frac{1}{(\delta_1+\varepsilon)T+\sqrt{T}}$ and every $T \geq 1$, the iterates $\boldsymbol{w}_t$ from (16) satisfy

$$
\mathbb{E}\left[\ell(\bar{\boldsymbol{w}}_T) - \inf \ell\right] = \mathcal{O}\left(\frac{1}{\sqrt{T}} + \delta_1 + \varepsilon\right),
$$

where $\bar{\boldsymbol{w}}_T := \frac{1}{T}\sum_{t=0}^{T-1} \theta^t \boldsymbol{w}_t$ with $\theta = \frac{T}{T+2}$.

The proofs follow techniques from the analysis of biased SGD (see e.g. Demidovich et al. (2023)) and are provided in Appendix A. Let us now discuss our main Assumption 5.1. Assume for simplicity that $\delta_2 = \sigma_2 = 0$ (for instance if $\ell$ is Lipschitz), that $\boldsymbol{e}_t = 0$ (the sample median is computed exactly), and consider that $\boldsymbol{g}_t^{(i)} = \nabla\ell(\boldsymbol{w}_t) + \boldsymbol{\xi}_t^{(i)}$, where $\boldsymbol{\xi}_t^{(i)}$ is a noise term. Then, Assumption 5.1 becomes[1]

$$
\mathbb{E}\left[\|\texttt{median}_{\ell_p, i \in [n]}(\boldsymbol{\xi}_t^{(i)})\|^2\right] \leq \nu^2 < +\infty.
\tag{18}
$$

In other words, we need the second moment of the sample median of $\boldsymbol{\xi}_t^{(i)}$ to be uniformly bounded. We want to stress that our choice of using the median is crucial when the $\boldsymbol{\xi}_t^{(i)}$ come from a heavy-tailed distribution, such as a (univariate) standard $\alpha$-stable distribution, with $\alpha \in (1, 2)$. Indeed, in this case we know that the variance of the sample median is *finite* for sufficiently large sample size $n$ (see Bickel (1967, Theorem 2.2) and Nolan (2020, Theorem 1.2))[2]. On the contrary, taking $\boldsymbol{m}_t$ as the usual sample *mean* causes trouble, as the variance of the sample mean is *infinite*[3] everywhere. See Fig. 2 for an illustration of this phenomenon.

This simple discussion illustrates that Assumption 5.1 can be satisfied under mild conditions on the noise. We exploit this in the next corollary, which shows that the $\ell_1$-sample median can even accommodate state-dependent noise. Let us define first our model for such noise.

---

[1]See Lemma A.2 in the appendix for a proof.

[2]We know more: The sample median of a parent distribution with median $\xi$ is asymptotically normal with location $\xi$ (Chu & Hotelling, 1955). For Cauchy distributions ($\alpha = 1$), the sample median has a finite variance for $n \geq 5$ (Chu & Hotelling, 1955, Remark 1), while first and second moment of the sample mean are undefined.

[3]This is due to the fact that, by definition of $\alpha$-stability, the sample mean of i.i.d sample follows the same $\alpha$-stable distribution (up to affine translation).

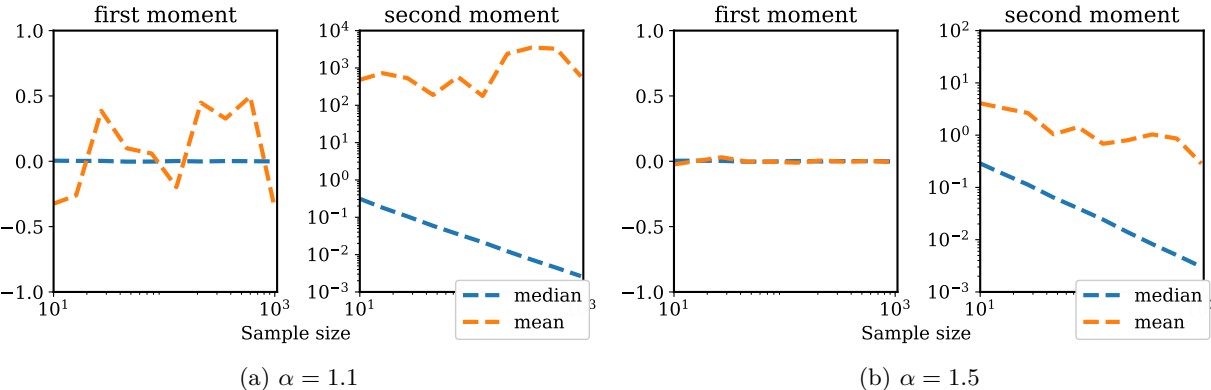

Figure 2: First and second moment of sample median/mean for standard $\alpha$-stable distribution. We approximate $\mathbb{E}$ by running $10\,000$ trials. The sample median has much smaller variance (cf. (18)) compared to the sample mean.

---

**Assumption 5.5.** For every $\boldsymbol{w} \in \mathbb{R}^d$, we have that $\boldsymbol{g} = \nabla\ell(\boldsymbol{w}) + \boldsymbol{\Sigma}(\boldsymbol{w})\boldsymbol{\zeta}$, where $\boldsymbol{\Sigma}(\boldsymbol{w}) \in \mathbb{R}^{d\times d}$ is a matrix and $\boldsymbol{\zeta}$ is a random vector whose components are drawn i.i.d. from a standard symmetric $\alpha$-stable distribution with $\alpha > 1$. There exists constants $c_1, c_2 > 0$ such that

$$\|\boldsymbol{\Sigma}(\boldsymbol{w})\|_F^2 \le c_1 + c_2\|\nabla\ell(\boldsymbol{w})\|^2 \quad \text{for all } \boldsymbol{w} \in \mathbb{R}^d.$$

---

In the recent literature (see e.g. Zhang et al. (2020b); Puchkin et al. (2023); Sadiev et al. (2023)), a commonly made assumption is that $\mathbb{E}\left[\|\boldsymbol{g} - \nabla\ell(\boldsymbol{w})\|^p\right] \le \sigma^2$ for some $p < 2$ where $\sigma^2 > 0$ cannot depend on $\boldsymbol{w}$. On the contrary, Assumption 5.5 can accommodate much stronger noise. To illustrate this, consider $d = 1$ and $\ell(\boldsymbol{w}) = \boldsymbol{w}^2/2$. Assumption 5.5 allows the noise model $\boldsymbol{\Sigma}(\boldsymbol{w})\boldsymbol{\zeta}$ with $\boldsymbol{\Sigma}(\boldsymbol{w}) = \sqrt{c_1 + c_2\boldsymbol{w}^2}$, which results in strong noise, since $\boldsymbol{\zeta}$ has infinite variance if $\alpha < 2$, and this is amplified by $\boldsymbol{\Sigma}(\boldsymbol{w})$ when $\boldsymbol{w}$ is large. Such strong noise makes vanilla `SGD` diverge, as we will illustrate in our experiments.

---

**Corollary 5.6.** Let $n \ge 3$ be odd and $p = 1$. Consider the iterates from (16) with $\boldsymbol{e}_t = 0$, and suppose that $\boldsymbol{g}_t^{(i)} = \nabla\ell(\boldsymbol{w}_t) + \boldsymbol{\Sigma}(\boldsymbol{w}_t)\boldsymbol{\zeta}_t^{(i)}$ verifies Assumption 5.5.

  (i) If $\ell$ is $L$-smooth and $\eta = \mathcal{O}(\frac{1}{\sqrt{T}})$, then we have that $\min_{0 \le t \le T-1} \mathbb{E}\left[\|\nabla\ell(\boldsymbol{w}_t)\|^2\right] = \mathcal{O}(1/\sqrt{T})$.

  (ii) If $\ell$ is convex and Lipschitz and $\eta = 1/\sqrt{T}$ then we have that $\mathbb{E}\left[\ell(\bar{\boldsymbol{w}}_T) - \inf\ell\right] = \mathcal{O}(1/\sqrt{T})$, where $\bar{\boldsymbol{w}}_T$ is defined as in Proposition 5.4.

---

This result illustrates how powerful the sample median can be when used in gradient-based optimization. However, `SMGD` (16) has clear drawbacks: (i) it requires to draw multiple samples per iteration (multiple calls of the oracle), and (ii) we need to compute the sample median up to some tolerance. For instance, the geometric median can not be computed in closed form and there exists a vast body of literature on how to approximate it (Cohen et al., 2016). Moreover, the noise still needs to be symmetric in order for the median to denoise correctly (cf. Assumption 5.5 and Fig. 8 (right)).

Nevertheless, our convergence guarantees for `SMGD` allow us to be optimistic that the strong performance in the heavy-tailed setting will also transfer to the online drop-in replacements (of (16)), which we had previously presented in Section 4. We investigate this in the following experiments section.

# 6 Experiments

We compare the iterative gradient estimation techniques for two experimental setups, namely (i) estimating a vector/gradient from a fixed distribution, and (ii) methods of the form (13), that is, iterative gradient estimation while simultaneously learning the weights.

We study both settings for synthetic data where we can control the level of heavy-tailedness, as well as for language modelling tasks with transformer models. We consider the same three language modeling tasks as studied in (Kunstner et al., 2023): an encoder-only transformer for the `PTB` dataset, a Transformer-XL model for the `WikiText-2` dataset, and fine-tuning a `DistillBERT` model for question-answering on the `SQuAD` dataset. We refer to Appendix C for details.

In the following, we use `VClip` to refer to vectorwise clipping (10) and `CClip` for componentwise clipping (11).

## 6.1 Gradient Estimation with Fixed Weights

**Synthetic data.** In the first experiment, we verify the hypothesis that the choice of $\mathcal{D}$ matters for problem (8); in particular, we show that estimating the median is more stable when the noise distribution is heavy-tailed.

*Setup.* We consider the problem of estimating a fixed vector $\hat{\boldsymbol{g}}$ from the oracle $\boldsymbol{g} \sim \hat{\boldsymbol{g}} + \boldsymbol{\xi}$ with $\boldsymbol{\xi} \sim \mathcal{P}$, under varying degree of heavy-tailedness of $\mathcal{P}$. For this purpose, we choose the standard, unskewed $\alpha$-stable distribution $\mathcal{P}_\alpha$. We generate $\hat{\boldsymbol{g}} \in \mathbb{R}^d$ with $d = 10$ where each component is generated i.i.d standard Gaussian. In each iteration, a sample $\boldsymbol{g}_t$ is generated as follows: each coordinate of $(\boldsymbol{g}_t)_i$ is (independently) sampled from $\mathcal{P}_\alpha$ with location $(\hat{\boldsymbol{g}})_i$ and varying values for $0 < \alpha \le 2$. Importantly, the median of $\mathcal{P}_\alpha$ is equal to the location (i.e. $\hat{\boldsymbol{g}}$) for all values of $\alpha$. Recall that for $\alpha \le 1$ the mean of $\mathcal{P}_\alpha$ is not defined, and otherwise equal to the median.

We run the `SPP` method (9) for $\mathcal{D} \in \{\frac{1}{2}\|\cdot\|^2, \|\cdot\|_2, \|\cdot\|_1, H_\mu\}$, with a fixed step size $\tau = 0.01$ and track the relative $\ell_2$-error $\frac{\|\boldsymbol{m}_t - \hat{\boldsymbol{g}}\|_2}{\|\hat{\boldsymbol{g}}\|_2}$. The corresponding methods are called momentum, `VClip`, `CClip`, and `Huber`, where we set $\mu = 1.345$ according to Huber (1981). For each method and distribution, we run 1000 iterations and 50 different seeds.

*Discussion.* From Fig. 3 we find that as the noise becomes more heavy-tailed (as $\alpha$ decreases), momentum fails to produce accurate estimates of $\hat{\boldsymbol{g}}$. On the other hand, both `VClip` and `CClip` are robust to heavy tails, as expected, as we showed that `VClip` and `CClip` are estimating the median. Notably `CClip` becomes more accurate (relatively) as $\alpha$ decreases. The Huber function produces results very similar to `VClip`, but slightly inferior. We refer to Fig. 8 for additional ablations on the choice of $\tau$ and skewed noise.

**Language modelling.** We also run momentum, `VClip` and `CClip` in order to estimate the full gradient for the language modeling tasks, where the weights are *fixed at initialization*. The setup is identical to the one described by Kunstner et al. (2023), in particular Figure 1 therein. We present the results in Appendix C.1.

## 6.2 Training with Online Gradient Estimation

We now present comparisons of various (online) gradient estimators when simultaneously training the weights, that is the distribution of gradients changes in each iteration.

**Least squares with heavy-tailed noise.** We consider the noise-perturbed least-squares problem $\min_{\boldsymbol{w} \in \mathbb{R}^d} \frac{1}{2}\|\boldsymbol{w}\|^2$ and use a gradient oracle given by $\boldsymbol{g} \sim \boldsymbol{w} + \boldsymbol{\xi} \in \mathbb{R}^d$, where $\boldsymbol{\xi} \sim \mathcal{P}$ is the noise term. We consider three setups:

(S1) The $d$ components of $\boldsymbol{\xi}$ are i.i.d. distributed according to a standard $\alpha$-stable distribution with location zero and $\alpha = 1.1$. Note that $\mathbb{E}_{\boldsymbol{\xi} \sim \mathcal{P}}[\boldsymbol{\xi}] = \boldsymbol{0}$.

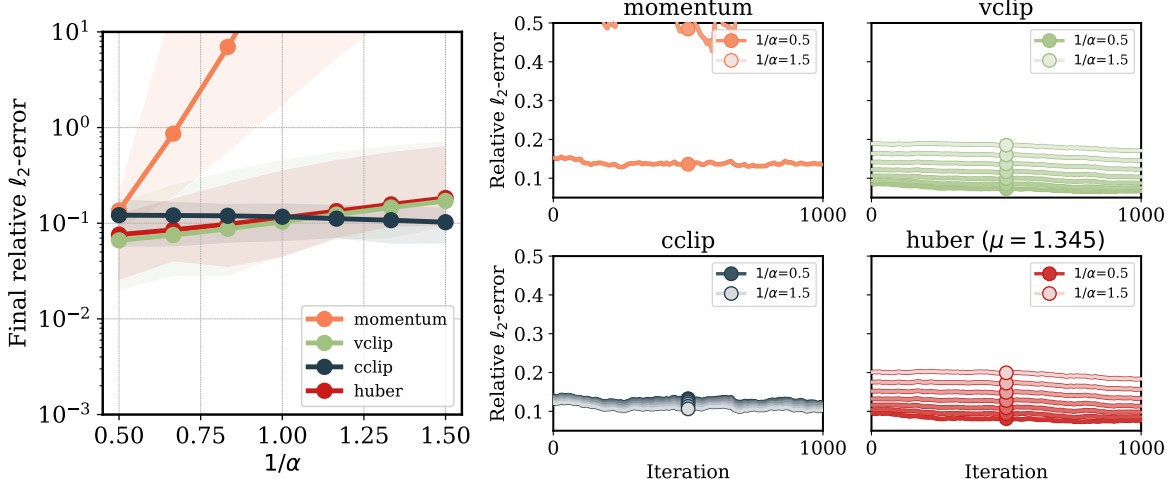

Figure 3: $\tau = 0.01$ **Left:** Final error for varying values of $\alpha$ (from left to right, distributions are more heavy-tailed). Shaded area marks minimal and maximal value over the 50 independent runs. **Right:** Convergence plot for all methods for $\frac{1}{\alpha} \in [0.5, 1.5]$ (higher value of $\frac{1}{\alpha}$ corresponds to heavier tails).

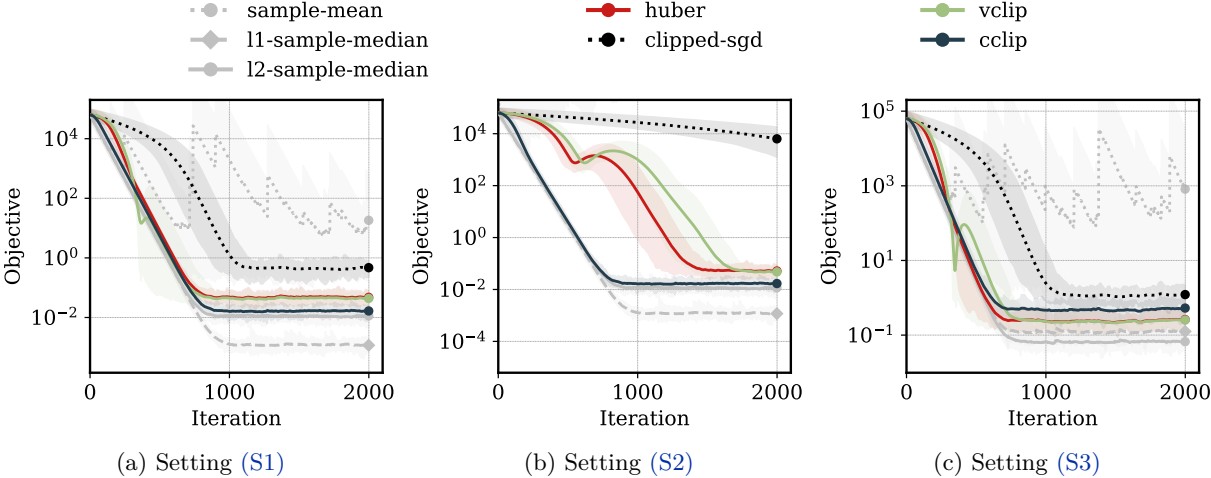

(a) Setting (S1)      (b) Setting (S2)      (c) Setting (S3)

Figure 4: Least squares with heavy-tailed noise that is independent over components (**left**), state-dependent (**middle**), and where components of noise are dependent (**right**). All methods in gray use $n = 5$ samples per iteration, all others only one. Note that for (S2), the sample mean **immediately diverges**, and hence does not appear in the plot. Shaded area depicts minimum and maximum value over 50 repetitions.

(S2) Same as (S1), but the noise is state-dependent, given by $\boldsymbol{\xi}_t = \sigma \cdot \boldsymbol{\xi}$ where $\sigma = \sqrt{1 + \|\boldsymbol{w}_t\|^2}$.

(S3) $\boldsymbol{\xi}$ follows an elliptically contoured $\alpha$-stable distribution (Nolan, 2013) with location zero and $\alpha = 1.1$. In particular, the components of $\boldsymbol{\xi}$ are not independent.

We run seven different methods: on the one hand, the method depicted in Section 5, (16), where in each iteration we draw $n = 5$ samples. We run three variations of (16), namely using the $\ell_1$- and $\ell_2$-sample median as well as the sample mean for $\boldsymbol{m}_t$. The $\ell_2$-median is approximately computed using the method proposed in Vardi & Zhang (2000).

On the other hand, we run our online median estimator methods (13), with $\mathcal{D} \in \{\|\cdot\|_2, \|\cdot\|_1, H_\mu\}$, that is, `VClip`, `CClip`, and `Huber`. Importantly, these methods only receive *one* sample per iteration, and they do not involve any median computations (in contrast to $\ell_1$- and $\ell_2$-sample median). We also run `clipped-SGD`

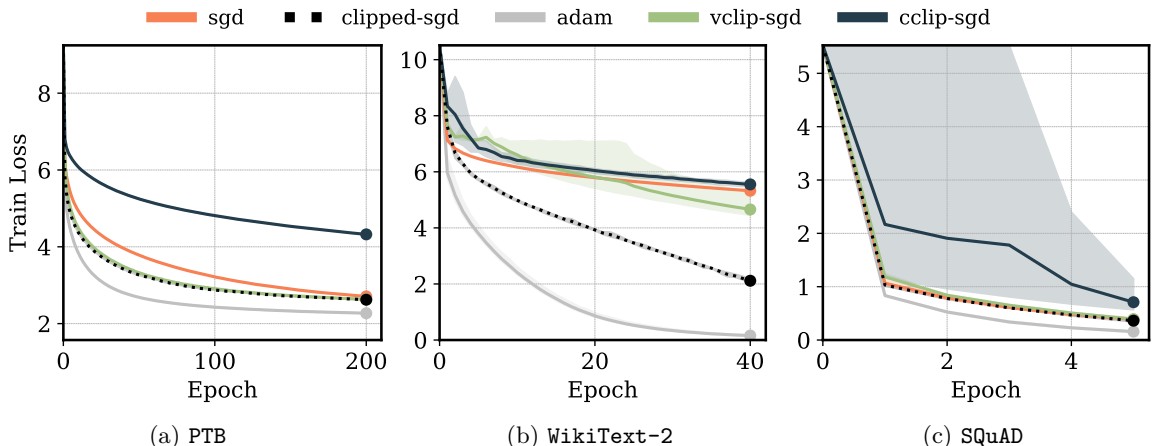

Figure 5: Training loss for each method, with tuned learning rate. Shaded area depicts minimum and maximum value over 5 seeds.

(cf. (14)) with $\beta = 0.9$ and $c = 50$.[4] We run all methods with a learning rate $\eta_t = 0.01$. For VClip, CClip, and Huber we set $\tau = 1$, and we set again $\mu = 1.345$ for Huber. We always run 50 repetitions and report averaged metrics.

*Discussion.* Fig. 4 shows that both $\ell_1$- and $\ell_2$-sample median attain the smallest objective, while the sample mean does not converge as predicted by our theory (see discussion after Proposition 5.3). From the online methods, CClip performs best in the scenario where the noise is componentwise independent, while VClip and Huber are better in the other case. Overall, Huber and VClip behave again very similarly (a more detailed analysis of Huber with different values of $\mu$ can be found in Fig. 9). We find that both VClip and CClip reach much smaller loss values than clipping the stochastic gradient directly, and thus clipping only the increment is better suited here. We also find that the $\ell_1$-sample median can deal with state-dependent noise (S2) as predicted by our theory.

**Language modeling.** We now consider the online methods (13), that is training while simultaneously estimating the gradient, for the three language modeling tasks. We compare SGD-M, VClip and CClip, and clipped-SGD.[5] We also add Adam as a baseline, but focus on the comparison among the SGD-type methods. For all methods, we tune the learning rate on a $\log_{10}$-scaled grid (tuned values reported in Table 1), displaying the one that attained smallest final training loss (averaged over 5 seeds).

We choose a standard momentum/clipping parameters for all tasks (without tuning): we set $\beta = 0.9$ for SGD-M, $\tau = 0.1$ for V/CClip, and $\beta = 0.9$, $c = 1$ for clipped-SGD.

*Discussion.* Fig. 5 shows that VClip improves over SGD-M for PTB and WikiText-2, and is on par for SQuAD. CClip performs always worse, which might be due to the noise being not componentwise independent. However, VClip does not close the gap to Adam, and also performs worse than clipped-SGD for the WikiText-2 dataset.

## 7 Conclusion

We provide a new approach for robust gradient estimation using the stochastic proximal point method, with a focus on (online) estimators of the median gradient. We observe that this general framework intersects with many existing works on robust stochastic optimization, where the connection between clipping and median estimation was not known previously.

---

[4]This choice is a tradeoff between slow convergence (large $c$) and high instability (small $c$).
[5]We also tried Huber (with $\mu = 1.3$) on these problems, but did not observe an advantage over VClip (plots not shown).

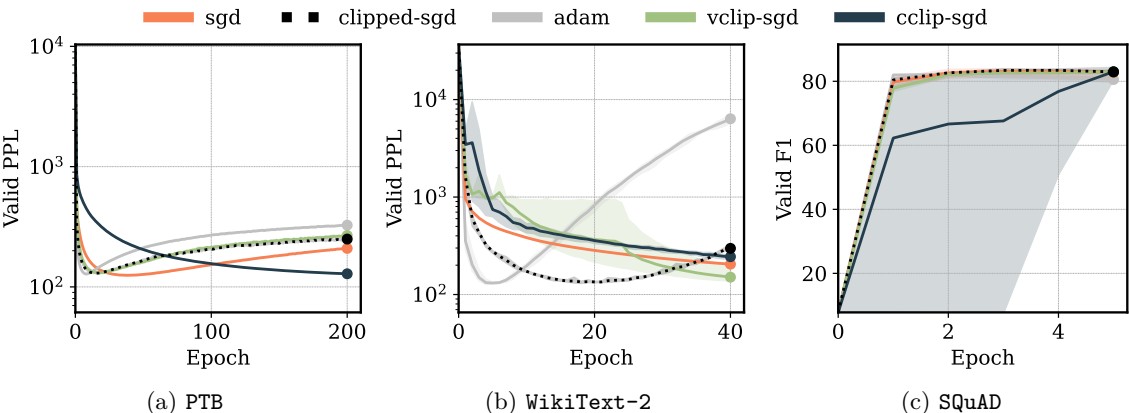

Figure 6: Validation score (measured as perplexity, or F1 score) for each method, with tuned learning rate. Shaded area depicts minimum and maximum value over 5 seeds.

We provide an analysis of `SGD` using the median of sampled gradients (called `SMGD`), and show convergence guarantees under potentially heavy-tailed and state-dependent noise. In contrast, under the same assumption, `SGD` with the sample mean does not converge.

Finally, numerical experiments show that online versions of `SMGD`, even though not covered by this theory, perform equally well on problems with heavy-tailed noise, while methods based on mean estimation fail. However, when training transformer models on language tasks, this gap is far less pronounced; this could indicate that the level of heavy-tailedness in these applications is moderate.

**Acknowledgments**

We thank the Scientific Computing Core at the Flatiron Institute, a division of the Simons Foundation, for the compute facilities and support. U.Ş. and F.S. are partially supported by the French government under the management of Agence Nationale de la Recherche as part of the "Investissements d'avenir" program, reference ANR-19-P3IA-0001 (PRAIRIE 3IA Institute), and the European Research Council Starting Grant DYNASTY – 101039676. G.G. gratefully acknowledges the Flatiron Institute for their hospitality and financial support during the visit when this work was written.

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

## Contents

## Appendix

## A  Missing Proofs for Section 5

### A.1  On the Assumptions on the Sample Median

We first state a simple fact about Lipschitz functions:

**Lemma A.1.** If Assumption 5.1 holds, and if $\ell$ is Lipschitz continuous, then without loss of generality we can assume that $\delta_2 = \sigma_2 = 0$.

*Proof.* If $\ell$ is $G$-Lipschitz, then for every subgradient $\boldsymbol{u} \in \partial\ell(\boldsymbol{w})$ we have $\|\boldsymbol{u}\| \leq G$, so one can replace $\delta_1^2$ with $\delta_1^2 + \delta_2 G^2$ and $\sigma_1^2$ with $\sigma_1^2 + \sigma_2 G^2$. □

In the following proof of Proposition 5.3 we will need an equivalent but more compact formulation of Assumption 5.1.

**Lemma A.2.** Let $\boldsymbol{m}$ be a random variable in $\mathbb{R}^d$, and consider the following property:

$$\mathbb{E}\left[\|\boldsymbol{m} - \nabla\ell(\boldsymbol{w})\|^2\right] \leq \nu_1^2 + \nu_2\|\nabla\ell(\boldsymbol{w})\|^2. \tag{19}$$

If (17) holds then (19) holds with $\nu_1^2 = \delta_1^2 + \sigma_1^2$ and $\nu_2 = \delta_2 + \sigma_2$. If (19) holds, then (17) holds with $\delta_1 = \sigma_1 = \nu_1$ and $\delta_2 = \sigma_2 = \nu_2$.

*Proof.* Expanding the squares, we get

$$\mathbb{E}\left[\|\boldsymbol{m} - \nabla\ell(\boldsymbol{w})\|^2\right] = \mathbb{E}\left[\|\boldsymbol{m} - \mathbb{E}\left[\boldsymbol{m}\right]\|^2\right] + \|\mathbb{E}\left[\boldsymbol{m}\right] - \nabla\ell(\boldsymbol{w})\|^2,$$

where we used that $\mathbb{E}\left[\langle \boldsymbol{m} - \mathbb{E}\left[\boldsymbol{m}\right], \mathbb{E}\left[\boldsymbol{m}\right] - \nabla\ell(\boldsymbol{w})\rangle\right] = 0$. Now, the first implication is trivial, and the second follows from the positivity of the involved terms. □

We now verify that Assumption 5.1 holds true for the $\ell_1$-sample median, provided that the noise on the subgradient is linearly depending on an $\alpha$-stable distribution, and that the linear coefficients grow with the norm of the subgradient.

**Lemma A.3.** Let $n \in \mathbb{N}$ be odd and $n \geq 3$. Suppose that the subgradient oracles $\boldsymbol{g}^{(i)} = \nabla\ell(\boldsymbol{w}) + \boldsymbol{\Sigma}(\boldsymbol{w})\boldsymbol{\zeta}^{(i)}$ verify Assumption 5.5, for all $i = 1, \ldots, n$ and set

$$\boldsymbol{m} = \mathtt{median}_{\ell_1, i \in [n]}\left(\boldsymbol{g}^{(i)}(\boldsymbol{w})\right).$$

Then, Assumption 5.1 holds with $\delta_1 = \delta_2 = 0$, and $\sigma_1^2 = C_n c_1$, $\sigma_2 = C_n c_2$, for some constant $C_n = \mathcal{O}(1/n)$.

*Proof.* Let us denote the noise vector by $\boldsymbol{\xi}^{(i)} = \boldsymbol{\Sigma}(\boldsymbol{w})\boldsymbol{\zeta}^{(i)}$. By the stability property of $\alpha$-stable distributions (Nolan, 2020, Proposition 1.3), we have that

$$\xi_k^{(i)} := \boldsymbol{\xi}_k^{(i)}(\boldsymbol{w}) = \sum_{j=1}^{d}[\boldsymbol{\Sigma}(\boldsymbol{w})]_{kj}\boldsymbol{\zeta}_j^{(i)} =^{\mathrm{d}} \|\boldsymbol{\Sigma}(\boldsymbol{w})_k\|_\alpha z, \tag{20}$$

where $=^{\mathrm{d}}$ denotes equality in distribution, $z$ is a standard symmetric $\alpha$-stable random variable in $\mathbb{R}$, and $\boldsymbol{\Sigma}(\boldsymbol{w})_k$ denotes the $k$-th row of $\boldsymbol{\Sigma}(\boldsymbol{w})$. Observe that from Assumption 5.5 we can write that

$$\sum_{i=1}^{d} \|\boldsymbol{\Sigma}(\boldsymbol{w})_i\|_\alpha^2 \leq \rho_\alpha^2 \sum_{i=1}^{d} \|\boldsymbol{\Sigma}(\boldsymbol{w})_i\|_2^2 = \rho_\alpha^2 \|\boldsymbol{\Sigma}(\boldsymbol{w})\|_F^2 \leq c_1' + c_2' \|\nabla \ell(\boldsymbol{w})\|^2, \quad \text{for all } \boldsymbol{w} \in \mathbb{R}^d. \qquad (21)$$

by setting $c_1' = \rho_\alpha^2 c_1$ and $c_2 = \rho_\alpha^2 c_2$, and where $\rho_\alpha > 0$ is the best constant such that $\| \cdot \|_\alpha \leq \rho_\alpha \| \cdot \|_2$, which is $\rho_\alpha = d^{\frac{2-\alpha}{2\alpha}}$ is for $\alpha \leq 2$, and $\rho_\alpha = 1$ otherwise.

We start by computing the bias term that is required by Assumption 5.1.

$$\begin{aligned}
\|\mathbb{E}\left[\boldsymbol{m}\right] - \nabla \ell(\boldsymbol{w})\|^2 =& \|\mathbb{E}\left[\texttt{median}_{\ell_1, i \in [n]} \left(\nabla \ell(\boldsymbol{w}) + \boldsymbol{\xi}^{(i)}(\boldsymbol{w})\right)\right] - \nabla \ell(\boldsymbol{w})\|^2 \\
=& \|\mathbb{E}\left[\texttt{median}_{\ell_1, i \in [n]} \left(\boldsymbol{\xi}^{(i)}(\boldsymbol{w})\right)\right]\|^2
\end{aligned}$$

where we used the fact that

$$\texttt{median}_{\ell_1, i \in [n]} \left(\nabla \ell(\boldsymbol{w}) + \boldsymbol{\xi}^{(i)}(\boldsymbol{w})\right) = \nabla \ell(\boldsymbol{w}) + \texttt{median}_{\ell_1, i \in [n]} \left(\boldsymbol{\xi}^{(i)}(\boldsymbol{w})\right).$$

On the other hand, the $\ell_1$-median boils down to computing component-wise medians, hence we have that

$$\mathbb{E}\left[\texttt{median}_{\ell_1, i \in [n]} \left(\boldsymbol{\xi}^{(i)}(\boldsymbol{w})\right)\right] = \begin{bmatrix} \mathbb{E}\left[\texttt{median}_{i \in [n]} \left(\xi_1^{(i)}\right)\right] \\ \vdots \\ \mathbb{E}\left[\texttt{median}_{i \in [n]} \left(\xi_d^{(i)}\right)\right] \end{bmatrix},$$

where $\texttt{median}_{i \in [n]}(\xi_k^{(i)}) = \underset{m \in \mathbb{R}}{\arg\min} \frac{1}{n} \sum_{i=1}^{n} |m - \xi_k^{(i)}|$ is the one dimensional sample median estimator.

Now, let us compute $\mathbb{E}\left[\texttt{median}_{i \in [n]} \left(\xi_k^{(i)}\right)\right]$ for some $k \in \{1, \ldots, d\}$. Denoting $r = (n-1)/2 \in \mathbb{N}$ and by using the probability density function of the sample median (cf. Maritz & Jarrett (1978)), we have that:

$$\mathbb{E}\left[\texttt{median}_{i \in [n]} \left(\xi_k^{(i)}\right)\right] = \frac{n!}{r!r!} \underbrace{\int_{\mathbb{R}} x \left(F_{\boldsymbol{w},k}(x)(1 - F_{\boldsymbol{w},k}(x))\right)^r p_{\boldsymbol{w},k}(x) \mathrm{d}x,}_{=:A}$$

where $F_{\boldsymbol{w},k}$ and $p_{\boldsymbol{w},k}(x)$ are respectively the cumulative distribution function (cdf) and the probability density function (pdf) of $\xi_k^{(i)} \sim \mathcal{P}_{\alpha, \|\boldsymbol{\Sigma}_k(\boldsymbol{w})\|_\alpha}$ that is a symmetric $\alpha$-stable random variable with scale $\|\boldsymbol{\Sigma}_k(\boldsymbol{w})\|_\alpha$. As $p_{\boldsymbol{w},k}$ is continuous and symmetric around zero, we have $p_{\boldsymbol{w},k}(x) = p_{\boldsymbol{w},k}(-x)$ and $(1 - F_{\boldsymbol{w},k}(x)) = F_{\boldsymbol{w},k}(-x)$ for all $x \in \mathbb{R}$. This gives us:

$$\begin{aligned}
A =& \int_0^\infty x \left(F_{\boldsymbol{w},k}(x) F_{\boldsymbol{w},k}(-x)\right)^r p_{\boldsymbol{w},k}(x) \mathrm{d}x + \int_{-\infty}^0 x \left(F_{\boldsymbol{w},k}(x) F_{\boldsymbol{w},k}(-x)\right)^r p_{\boldsymbol{w},k}(x) \mathrm{d}x \\
=& \int_0^\infty x \left(F_{\boldsymbol{w},k}(x) F_{\boldsymbol{w},k}(-x)\right)^r p_{\boldsymbol{w},k}(x) \mathrm{d}x - \int_0^\infty x \left(F_{\boldsymbol{w},k}(x) F_{\boldsymbol{w},k}(-x)\right)^r p_{\boldsymbol{w},k}(x) \mathrm{d}x \\
=& 0.
\end{aligned}$$

Hence, we obtain that

$$\mathbb{E}\left[\texttt{median}_{\ell_1, i \in [n]} \left(\boldsymbol{\xi}^{(i)}(\boldsymbol{w})\right)\right] = 0.$$

This shows that the first line of Assumption 5.1 holds with $\delta_1 = \delta_2 = 0$.

We now proceed to the estimation of the variance. By using similar arguments, we have that:

$$
\begin{aligned}
\mathbb{E}\left[\|\boldsymbol{m} - \mathbb{E}\left[\boldsymbol{m}\right]\|^2\right] &= \mathbb{E}\left[\|\texttt{median}_{\ell_1, i \in [n]}\left(\boldsymbol{\xi}^{(i)}(\boldsymbol{w})\right)\|^2\right] \\
&= \sum_{k=1}^{d} \mathbb{E}\left[\left(\texttt{median}_{i \in [n]}\left(\xi_k^{(i)}\right)\right)^2\right] \\
&= \sum_{k=1}^{d} \frac{n!}{r!r!}\mathbb{E}\left[\left(\xi_k^{(i)}\right)^2\left(F_{\boldsymbol{w},k}\left(\xi_k^{(i)}\right)\left(1 - F_{\boldsymbol{w},k}\left(\xi_k^{(i)}\right)\right)\right)^r\right],
\end{aligned} \tag{22}
$$

where (22) follows from Maritz & Jarrett (1978, Equation 2.1). By using (20) with (21) and denoting the cdf of the standard symmetric $\alpha$-stable distribution by $F_z$, we further compute that:

$$
\begin{aligned}
\mathbb{E}\left[\|\boldsymbol{m} - \mathbb{E}\left[\boldsymbol{m}\right]\|^2\right] &= \frac{n!}{r!r!}\mathbb{E}\left[z^2\left(F_z(z)(1 - F_z(z))\right)^r\right]\sum_{k=1}^{d}\|\boldsymbol{\Sigma}_k(\boldsymbol{w})\|_\alpha^2 \\
&\leq \frac{n!}{r!r!}\mathbb{E}\left[z^2\left(F_z(z)(1 - F_z(z))\right)^r\right]\left(c_1' + c_2'\|\nabla\ell(\boldsymbol{w})\|^2\right) \\
&=: C_n\left(c_1' + c_2'\|\nabla\ell(\boldsymbol{w})\|^2\right).
\end{aligned}
$$

This shows that the second line of Assumption 5.1 holds with $\sigma_1^2 = C_n\rho_\alpha^2 c_1$ and $\sigma_2 = C_n\rho_\alpha^2 c_2$.

Finally we observe that, again by Maritz & Jarrett (1978, Equation 2.1), $C_n$ is the variance of the sample median of a set of i.i.d. standard symmetric $\alpha$-stable variables $\{z_1, \ldots, z_n\}$, where $z_i =^{\mathrm{d}} z$. As $n \to \infty$, it is well-known that the sample median is asymptotically normal with variance $1/(4nf^2(0))$, where $f$ denotes the pdf of $z$ (see Maritz & Jarrett (1978)). Hence we conclude that $C_n = \mathcal{O}(1/n)$ as $n \to \infty$. This concludes the proof. $\qquad\square$

## A.2 Rates for `SMGD` for Smooth Nonconvex Functions: Proof of Proposition 5.3

The following proposition is an application of biased `SGD` results to the `SMGD` algorithm with errors. It is possible to show that Assumptions 5.1 and 5.2 both imply Assumption 9 from Demidovich et al. (2023). So from a qualitative point of view the result below is the same that what we would obtain by using Theorem 3 from Demidovich et al. (2023). But doing so would produce a bound with worse constants, so we prefer to provide a proof exploiting directly our assumptions.

> **Proposition 5.3.** Let $\ell : \mathbb{R}^d \to \mathbb{R}$ be $L$-smooth and let Assumption 5.1 hold with $\delta_2 \leq \frac{1}{8}$. Take $\eta \leq \frac{1}{8L(1+2\sigma_2+2\delta_2)}$ and consider $\boldsymbol{w}_t$ a sequence generated by (16) where $\boldsymbol{e}_t$ verifies Assumption 5.2. Then, for every $T \geq 1$ it holds
> $$\min_{0 \leq t \leq T-1} \mathbb{E}\left[\|\nabla\ell(\boldsymbol{w}_t)\|^2\right] = \mathcal{O}\left(\frac{1}{\eta T} + \eta + \delta_1^2 + \varepsilon^2\right).$$

*Proof.* In this proof we note $\mathbb{E}_t$ for the expectation taken conditionally to the filtration generated by $\boldsymbol{w}_0, \ldots \boldsymbol{w}_t$. In particular, we will have from Assumption 5.2 on the errors together with the tower rule that $\mathbb{E}_t\|\boldsymbol{e}_t\|^2 = \mathbb{E}_t\mathbb{E}\left[\|\boldsymbol{e}_t\|^2 \mid \boldsymbol{g}_t^{(1)}, \ldots, \boldsymbol{g}_t^{(n)}\right] \leq \varepsilon^2$.

Smoothness of $\ell$ implies

$$
\ell(\boldsymbol{w}_{t+1}) - \ell(\boldsymbol{w}_t) - \langle\nabla\ell(\boldsymbol{w}_t), \boldsymbol{w}_{t+1} - \boldsymbol{w}_t\rangle \leq \frac{L}{2}\|\boldsymbol{w}_{t+1} - \boldsymbol{w}_t\|^2.
$$

If we note $\boldsymbol{d}_t := \boldsymbol{m}_t + \boldsymbol{e}_t$, thus $\boldsymbol{w}_{t+1} - \boldsymbol{w}_t = -\eta\boldsymbol{d}_t$. This implies

$$
\ell(\boldsymbol{w}_{t+1}) - \ell(\boldsymbol{w}_t) + \eta\langle\nabla\ell(\boldsymbol{w}_t), \boldsymbol{d}_t\rangle \leq \frac{L\eta^2}{2}\|\boldsymbol{d}_t\|^2.
$$

Let us bound the two terms depending on $\boldsymbol{d}_t$. First, take expectation conditioned to $\mathcal{F}_t$ (which we will denote $\mathbb{E}_t$) to write

$$
\begin{aligned}
\mathbb{E}_t\left[\langle \nabla\ell(\boldsymbol{w}_t), \boldsymbol{d}_t\rangle\right] &= \langle \nabla\ell(\boldsymbol{w}_t), \mathbb{E}_t\left[\boldsymbol{d}_t\right]\rangle = \frac{1}{2}\|\nabla\ell(\boldsymbol{w}_t)\|^2 + \frac{1}{2}\|\mathbb{E}_t\left[\boldsymbol{d}_t\right]\|^2 - \frac{1}{2}\|\mathbb{E}_t\left[\boldsymbol{d}_t\right] - \nabla\ell(\boldsymbol{w}_t)\|^2 \\
&\geq \frac{1}{2}\|\nabla\ell(\boldsymbol{w}_t)\|^2 - \frac{1}{2}\|\mathbb{E}_t\left[\boldsymbol{d}_t\right] - \nabla\ell(\boldsymbol{w}_t)\|^2.
\end{aligned}
$$

Using (17) and $\|\mathbb{E}_t\left[\boldsymbol{e}_t\right]\|^2 \leq \mathbb{E}_t\left[\|\boldsymbol{e}_t\|^2\right] \leq \varepsilon^2$ due to Jensen's inequality, we can bound

$$
\|\mathbb{E}_t\left[\boldsymbol{d}_t\right] - \nabla\ell(\boldsymbol{w}_t)\|^2 \leq 2\|\mathbb{E}_t\left[\boldsymbol{m}_t\right] - \nabla\ell(\boldsymbol{w}_t)\|^2 + 2\|\mathbb{E}_t\left[\boldsymbol{e}_t\right]\|^2 \leq 2\delta_1^2 + 2\delta_2\|\nabla\ell(\boldsymbol{w}_t)\|^2 + 2\varepsilon^2.
$$

Second, we use Young's inequality and our assumptions to bound $\|\boldsymbol{d}_t\|^2$:

$$
\begin{aligned}
\mathbb{E}_t\left[\|\boldsymbol{d}_t\|^2\right] &\leq 2\mathbb{E}_t\left[\|\boldsymbol{e}_t + \boldsymbol{m}_t - \nabla\ell(\boldsymbol{w}_t)\|^2\right] + 2\|\nabla\ell(\boldsymbol{w}_t)\|^2 \\
&\leq 4\mathbb{E}_t\left[\|\boldsymbol{e}_t\|^2\right] + 4\mathbb{E}_t\left[\|\boldsymbol{m}_t - \nabla\ell(\boldsymbol{w}_t)\|^2\right] + 2\|\nabla\ell(\boldsymbol{w}_t)\|^2 \\
&\leq 4\varepsilon^2 + 4(\delta_1^2 + \sigma_1^2) + (2 + 4\delta_2 + 4\sigma_2)\|\nabla\ell(\boldsymbol{w}_t)\|^2,
\end{aligned}
$$

where in the last inequality we used that (17) implies (19) (cf. Lemma A.2). Combine all the above inequalities to get

$$
\begin{aligned}
&\mathbb{E}_t\left[\ell(\boldsymbol{w}_{t+1})\right] - \ell(\boldsymbol{w}_t) + \frac{\eta}{2}\|\nabla\ell(\boldsymbol{w}_t)\|^2 - \frac{\eta}{2}(2\delta_1^2 + 2\delta_2\|\nabla\ell(\boldsymbol{w}_t)\|^2 + \varepsilon^2) \\
&\leq \frac{L\eta^2}{2}\left(4(\varepsilon^2 + \delta_1^2 + \sigma_1^2) + (2 + 4\delta_2 + 4\sigma_2)\|\nabla\ell(\boldsymbol{w}_t)\|^2\right).
\end{aligned}
$$

Rearranging yields

$$
\begin{aligned}
&\mathbb{E}_t\left[\ell(\boldsymbol{w}_{t+1})\right] - \ell(\boldsymbol{w}_t) \\
&\leq -\eta\left[\frac{1}{2} - \delta_2 - L\eta(1 + 2\sigma_2 + 2\delta_2)\right]\|\nabla\ell(\boldsymbol{w}_t)\|^2 + \eta(\delta_1^2 + \varepsilon^2) + 2L\eta^2(\varepsilon^2 + \delta_1^2 + \sigma_1^2).
\end{aligned}
$$

If $\delta_2 \leq \frac{1}{8}$ and using $\eta \leq \frac{1}{8L(1 + 2\sigma_2 + 2\delta_2)}$, after taking full expectation, we obtain

$$
\mathbb{E}\left[\ell(\boldsymbol{w}_{t+1})\right] - \mathbb{E}\left[\ell(\boldsymbol{w}_t)\right] \leq -\frac{\eta}{4}\mathbb{E}\left[\|\nabla\ell(\boldsymbol{w}_t)\|^2\right] + \eta(\delta_1^2 + \varepsilon^2) + 2L\eta^2(\varepsilon^2 + \delta_1^2 + \sigma_1^2).
$$

Sum over $t = 0, \ldots, T-1$, multiply by $\frac{4}{\eta T}$ and use $\min_{t=0,\ldots,T-1}\mathbb{E}\left[\|\nabla\ell(\boldsymbol{w}_t)\|^2\right] \leq \frac{1}{T}\sum_{t=0}^{T-1}\mathbb{E}\left[\|\nabla\ell(\boldsymbol{w}_t)\|^2\right]$ to finally obtain

$$
\min_{t=0,\ldots,T-1}\mathbb{E}\left[\|\nabla\ell(\boldsymbol{w}_t)\|^2\right] \leq \frac{4(\ell(\boldsymbol{w}_0) - \inf\ell)}{\eta T} + 8L\eta(\varepsilon^2 + \delta_1^2 + \sigma_1^2) + 4(\delta_1^2 + \varepsilon^2).
$$

$\square$

## A.3 Rates for Biased `SGD` for Convex Lipschitz Functions

In this section we prove another result about the complexity of biased `SGD`, that is `SGD` with biased estimators of the subgradient. As far as we know, biased `SGD` was only studied in the case of strongly convex functions or smooth nonconvex functions (Demidovich et al., 2023), or in the particular case of random projections. In that regard, the next result is a new contribution to the analysis of biased `SGD`.

Before stating our result, we recall a few standard definitions given that we are working with a nonsmooth convex function. Given a convex function $\ell : \mathbb{R}^d \to \mathbb{R}$, we say that $\boldsymbol{u}$ is a subgradient of $\ell$ at $\boldsymbol{w}$ if

$$
(\forall \boldsymbol{w}' \in \mathbb{R}^d) \quad \ell(\boldsymbol{w}') - \ell(\boldsymbol{w}) - \langle \boldsymbol{u}, \boldsymbol{w}' - \boldsymbol{w}\rangle \geq 0.
$$

We note $\partial\ell(\boldsymbol{w})$ the set of all subgradients of $\ell$ at $\boldsymbol{w}$, that is called the subdifferential of $\ell$ at $\boldsymbol{w}$.

**Theorem A.4.** Let $\ell : \mathbb{R}^d \to \mathbb{R}$ be a convex $G$-Lipschitz function such that $\operatorname{argmin} \ell \neq \emptyset$. Let $\eta > 0$, and note $D := \|\boldsymbol{w}_0 - \boldsymbol{w}_*\|$ for some $\boldsymbol{w}_* \in \operatorname{argmin} \ell$. Consider the iterates generated by a biased Stochastic Subgradient Descent method

$$\boldsymbol{w}_{t+1} = \boldsymbol{w}_t - \eta \boldsymbol{g}_t,$$

and assume that the estimator $\boldsymbol{g}_t$ has a uniformly finite bias and variance (we denote by $\mathcal{F}_t$ the filtration induced by $\boldsymbol{w}_0, \ldots, \boldsymbol{w}_t$):

$$\|\mathbb{E}\left[\boldsymbol{g}_t | \mathcal{F}_t\right] - \boldsymbol{u}_t\| \leq \delta, \quad \mathbb{E}\left[\|\boldsymbol{g}_t - \mathbb{E}\left[\boldsymbol{g}_t | \mathcal{F}_t\right]\|^2 | \mathcal{F}_t\right] \leq \sigma^2,$$

where $\boldsymbol{u}_t$ is any subgradient of $\ell$ at $\boldsymbol{w}_t$. Then, for every $T \geq 1$

$$\mathbb{E}\left[\ell(\bar{\boldsymbol{w}}_T) - \inf \ell\right] \leq \frac{2D^2}{\eta T} + \eta \frac{\sigma^2 + (\delta + G)^2}{2} + \eta \frac{\delta^2 T}{4}$$

where $\bar{\boldsymbol{w}}_T$ is an average of the first $T$ iterates defined by $\bar{\boldsymbol{w}}_T = \frac{1}{\sum_{t=0}^{T-1} \theta^t} \sum_{t=0}^{T-1} \theta^t \boldsymbol{w}_t$, $\theta = \frac{T}{T+2}$. In particular, if we take a constant step size equal to

$$\eta = \frac{2\sqrt{2}D}{\sqrt{2(\sigma^2 + (\delta + G)^2)T + \delta^2 T^2}} \quad (\text{resp. } \eta = \frac{1}{\delta T + \sqrt{T}} )$$

then we can guarantee that

$$\mathbb{E}\left[\ell(\bar{\boldsymbol{w}}_T) - \inf \ell\right] \leq 2D\sqrt{\frac{\sigma^2 + (\delta + G)^2}{T} + \frac{\delta^2}{2}} \quad (\text{resp. } \frac{2D^2 + \sigma^2 + \delta^2 + G^2}{\sqrt{T}} + (2D^2 + 1)\delta ).$$

*Proof.* Let $w_* \in \operatorname{argmin} \ell$ and $T \geq 1$ be fixed. Develop the squares and use the definition of the algorithm to write

$$
\begin{aligned}
\|\boldsymbol{w}_{t+1} - \boldsymbol{w}_*\|^2 &= \|\boldsymbol{w}_t - \boldsymbol{w}_*\|^2 + 2\langle \boldsymbol{w}_{t+1} - \boldsymbol{w}_t, \boldsymbol{w}_t - \boldsymbol{w}_* \rangle + \|\boldsymbol{w}_{t+1} - \boldsymbol{w}_t\|^2 \\
&= \|\boldsymbol{w}_t - \boldsymbol{w}_*\|^2 - 2\eta\langle \boldsymbol{g}_t, \boldsymbol{w}_t - \boldsymbol{w}_* \rangle + \eta^2 \|\boldsymbol{g}_t\|^2.
\end{aligned}
$$

Let $\bar{\boldsymbol{g}}_t$ be the projection of $\mathbb{E}\left[\boldsymbol{g}_t | \mathcal{F}_t\right]$ onto $\partial \ell(\boldsymbol{w}_t)$, the subdifferential of $\ell$ at $\boldsymbol{w}_t$, which is a nonempty closed convex set. This means that from our assumption on the bias of $\boldsymbol{g}_t$ we will have

$$\|\mathbb{E}\left[\boldsymbol{g}_t | \mathcal{F}_t\right] - \bar{\boldsymbol{g}}_t\| \leq \|\mathbb{E}\left[\boldsymbol{g}_t | \mathcal{F}_t\right] - \boldsymbol{u}_t\| \leq \delta.$$

Then, after taking conditional expectation (denoted by $\mathbb{E}_t\left[\cdot\right]$ instead of $\mathbb{E}\left[\cdot | \mathcal{F}_t\right]$) we can write

$$
\begin{aligned}
\mathbb{E}_t\left[\|\boldsymbol{w}_{t+1} - \boldsymbol{w}_*\|^2\right] &= \|\boldsymbol{w}_t - \boldsymbol{w}_*\|^2 - 2\eta\langle \mathbb{E}_t\left[\boldsymbol{g}_t\right], \boldsymbol{w}_t - \boldsymbol{w}_* \rangle + \eta^2 \mathbb{E}_t\left[\|\boldsymbol{g}_t\|^2\right] \\
&= \|\boldsymbol{w}_t - \boldsymbol{w}_*\|^2 - 2\eta\langle \bar{\boldsymbol{g}}_t, \boldsymbol{w}_t - \boldsymbol{w}_* \rangle + 2\eta\langle \bar{\boldsymbol{g}}_t - \mathbb{E}_t\left[\boldsymbol{g}_t\right], \boldsymbol{w}_t - \boldsymbol{w}_* \rangle + \eta^2 \mathbb{E}_t\left[\|\boldsymbol{g}_t\|^2\right] \\
&\leq \|\boldsymbol{w}_t - \boldsymbol{w}_*\|^2 - 2\eta(\ell(\boldsymbol{w}_t) - \inf \ell) + 2\eta\langle \bar{\boldsymbol{g}}_t - \mathbb{E}_t\left[\boldsymbol{g}_t\right], \boldsymbol{w}_t - \boldsymbol{w}_* \rangle + \eta^2 \mathbb{E}_t\left[\|\boldsymbol{g}_t\|^2\right],
\end{aligned}
$$

where in the last inequality we used the convexity of $\ell$ together with the fact that $\bar{\boldsymbol{g}}_t \in \partial \ell(\boldsymbol{w}_t)$. The last term can be bounded by using our assumptions on the estimator $\boldsymbol{g}_t$, and the fact that $\ell$ has bounded subgradients:

$$\mathbb{E}_t\left[\|\boldsymbol{g}_t\|^2\right] = \mathbb{E}_t\left[\|\boldsymbol{g}_t - \mathbb{E}_t\left[\boldsymbol{g}_t\right]\|^2\right] + \|\mathbb{E}_t\left[\boldsymbol{g}_t\right]\|^2 \leq \sigma^2 + (\|\mathbb{E}_t\left[\boldsymbol{g}_t\right] - \bar{\boldsymbol{g}}_t\| + \|\bar{\boldsymbol{g}}_t\|)^2 \leq \sigma^2 + (\delta + G)^2.$$

Injecting this in the previous inequality we obtain

$$\mathbb{E}_t\left[\|\boldsymbol{w}_{t+1} - \boldsymbol{w}_*\|^2\right] \leq \|\boldsymbol{w}_t - \boldsymbol{w}_*\|^2 - 2\eta(\ell(\boldsymbol{w}_t) - \inf \ell) + 2\eta\langle \bar{\boldsymbol{g}}_t - \mathbb{E}_t\left[\boldsymbol{g}_t\right], \boldsymbol{w}_t - \boldsymbol{w}_* \rangle + \eta^2\sigma^2 + \eta^2(\delta + G)^2.$$

Now we introduce a parameter $\varepsilon > 0$, and we use Young's inequality together with our assumption on the bias of $\boldsymbol{g}_t$ to write

$$2\langle \bar{\boldsymbol{g}}_t - \mathbb{E}_t\left[\boldsymbol{g}_t\right], \boldsymbol{w}_t - \boldsymbol{w}_* \rangle \leq \frac{1}{\varepsilon}\|\bar{\boldsymbol{g}}_t - \mathbb{E}_t\left[\boldsymbol{g}_t\right]\|^2 + \varepsilon\|\boldsymbol{w}_t - \boldsymbol{w}_*\|^2 \leq \varepsilon^{-1}\delta^2 + \varepsilon\|\boldsymbol{w}_t - \boldsymbol{w}_*\|^2.$$

Inject this bound in our main inequality

$$\mathbb{E}_t \left[\|\boldsymbol{w}_{t+1} - \boldsymbol{w}_*\|^2\right] \leq (1 + \eta\varepsilon)\|\boldsymbol{w}_t - \boldsymbol{w}_*\|^2 - 2\eta(\ell(\boldsymbol{w}_t) - \inf \ell) + \eta^2\sigma^2 + \eta^2(\delta + G)^2 + \eta\varepsilon^{-1}\delta^2,$$

and after reorganizing the terms, dividing by $2\eta$ and taking expectation we finally obtain

$$\mathbb{E}\left[\ell(\boldsymbol{w}_t) - \inf \ell\right] \leq \frac{1 + \eta\varepsilon}{2\eta}\mathbb{E}\left[\|\boldsymbol{w}_t - \boldsymbol{w}_*\|^2\right] - \frac{1}{2\eta}\mathbb{E}\left[\|\boldsymbol{w}_{t+1} - \boldsymbol{w}_*\|^2\right] + \eta\frac{\sigma^2 + (\delta + G)^2}{2} + \varepsilon^{-1}\frac{\delta^2}{2}.$$

We are now ready to use a weighted telescopic sum argument. First, define $\theta := \frac{1}{1+\eta\varepsilon}$ and $\rho_t := \theta^t$. Second, multiply our inequality by $\rho_{t+1}$ and then sum for $t = 0, \ldots, T-1$. Observe that the term $(1+\eta\varepsilon)\rho_{t+1}$ is equal to $\rho_t$ due to our definition, which means that we have a telescoping sum where the terms $\frac{\rho_t}{2\eta}\mathbb{E}\left[\|\boldsymbol{w}_t - \boldsymbol{w}_*\|^2\right]$ will cancel each other. Third, divide by $\sum_{t=0}^{T-1}\rho_{t+1}$ so to obtain

$$\frac{1}{\sum_{t=0}^{T-1}\rho_{t+1}} \sum_{t=0}^{T-1} \rho_{t+1}\mathbb{E}\left[\ell(\boldsymbol{w}_t) - \inf \ell\right] \leq \frac{\rho_0}{2\eta\sum_{t=0}^{T-1}\rho_{t+1}}\mathbb{E}\left[\|\boldsymbol{w}_0 - \boldsymbol{w}_*\|^2\right] + \eta\frac{\sigma^2 + (\delta + G)^2}{2} + \varepsilon^{-1}\frac{\delta^2}{2}.$$

Now define $\bar{\boldsymbol{w}}_T := \frac{1}{\sum_{t=0}^{T-1}\rho_{t+1}} \sum_{t=0}^{T-1} \rho_{t+1}\boldsymbol{w}_t = \frac{1}{\sum_{t=0}^{T-1}\theta^t} \sum_{t=0}^{T-1} \theta^t\boldsymbol{w}_t$, and use Jensen's inequality to get

$$\mathbb{E}\left[\ell(\bar{\boldsymbol{w}}_T) - \inf \ell\right] \leq \frac{\|\boldsymbol{w}_0 - \boldsymbol{w}_*\|^2}{2\eta\sum_{t=0}^{T-1}\rho_{t+1}} + \eta\frac{\sigma^2 + (\delta + G)^2}{2} + \varepsilon^{-1}\frac{\delta^2}{2}.$$

We will now simplify this upper bound, so we can later make an appropriate choice of $\eta$ leading to the desired complexity bound. We will now focus on the geometric sum appearing in the denominator, and lower bound it. Start by using the definition of $\rho_t$ to write

$$\eta\sum_{t=0}^{T-1} \rho_{t+1} = \eta\theta\sum_{t=0}^{T-1} \theta^t = \eta\frac{\theta}{1-\theta}(1 - \theta^T).$$

On the one hand, we see that from the definition of $\theta$ we have

$$\eta\frac{\theta}{1-\theta} = \frac{\eta}{1+\eta\varepsilon}\frac{1}{1 - \frac{1}{1+\eta\varepsilon}} = \frac{\eta}{\eta\varepsilon} = \varepsilon^{-1}.$$

On the other hand, we can guarantee that $(1 - \theta^T) \geq 1/2$ provided that $\varepsilon = \frac{2}{\eta T}$. To see this, first observe that $(1 - \theta^T) \geq 1/2$ is equivalent to $T \geq \frac{\ln(2)}{\ln(\theta^{-1})}$. Since we impose that $\eta\varepsilon = \frac{2}{T}$ then necessarily $\theta = \frac{T}{T+2}$ which means that $\theta^{-1} = 1 + \frac{2}{T}$. So what we need to verify is $T \geq \frac{\ln(2)}{\ln(1+\frac{2}{T})}$. Now, observe that $\ln(2) \leq 1$. Moreover, it is an exercise (see Lemma A.5 which is deferred to the end of this proof) to verify that for all $x \geq 0$, $\frac{1}{\ln(x+1)} \leq \frac{1}{2} + \frac{1}{x}$. So for our condition to hold it is enough to ask that $T \geq \frac{1}{2} + \frac{T}{2}$, which is equivalent to $T \geq 1$, which is true.

Combining all those bounds together with our new definition for $\varepsilon$, we now have

$$\mathbb{E}\left[\ell(\bar{\boldsymbol{w}}_T) - \inf \ell\right] \leq \frac{2\|\boldsymbol{w}_0 - \boldsymbol{w}_*\|^2}{\eta T} + \eta\frac{\sigma^2 + (\delta + G)^2}{2} + \frac{\eta\delta^2 T}{4}.$$

To simplify the last stage of our analysis, let us note $a = 2\|\boldsymbol{w}_0 - \boldsymbol{w}_*\|^2$, $b = \frac{\sigma^2 + (\delta + G)^2}{2}$ and $c = \frac{\delta^2}{4}$, so that our bound writes as

$$\mathbb{E}\left[\ell(\bar{\boldsymbol{w}}_T) - \inf \ell\right] \leq \frac{a}{\eta T} + \eta b + c\eta T. \tag{23}$$

Minimizing the right-hand side with respect to $\eta$ is equivalent to solve $\frac{a}{\eta T} = \eta b + c\eta T$, where

$$\frac{a}{\eta T} = \eta b + c\eta T \Leftrightarrow \eta^2 = \frac{a}{T(b + cT)} \Leftrightarrow \eta = \sqrt{\frac{a}{bT + cT^2}} \Leftrightarrow \eta = 2\sqrt{\frac{2\|\boldsymbol{w}_0 - \boldsymbol{w}_*\|^2}{2(\sigma^2 + (\delta + G)^2)T + \delta^2 T^2}}$$

With such choice of stepsize, our bound becomes

$$\mathbb{E}\left[\ell(\bar{\boldsymbol{w}}_T) - \inf \ell\right] \leq \frac{2a}{\eta T} = \frac{2a}{T}\sqrt{\frac{bT + cT^2}{a}} = 2\sqrt{a}\sqrt{\frac{b}{T} + c} = 2\|\boldsymbol{w}_0 - \boldsymbol{w}_*\|\sqrt{\frac{\sigma^2 + (\delta + G)^2}{T} + \frac{\delta^2}{2}}$$

and this gives us our first bound. For the second bound, let us take simply $\eta = \frac{1}{\delta T + \sqrt{T}}$ and inject it into (23) to obtain

$$\begin{aligned}
\mathbb{E}\left[\ell(\bar{\boldsymbol{w}}_T) - \inf \ell\right] &\leq \frac{a\delta T + a\sqrt{T}}{T} + \frac{b}{\delta T + \sqrt{T}} + \frac{cT}{\delta T + \sqrt{T}} \\
&\leq a\delta + \frac{a}{\sqrt{T}} + \frac{b}{\sqrt{T}} + \frac{c}{\delta} = \frac{a+b}{\sqrt{T}} + a\delta + \frac{c}{\delta} \\
&\leq \frac{2D^2 + \sigma^2 + \delta^2 + G^2}{\sqrt{T}} + (2D^2 + 1)\delta,
\end{aligned}$$

where in the last inequality we simplified some numerical constants. □

**Lemma A.5.** For every $x \geq 0$, $\frac{1}{\ln(1+x)} \leq \frac{1}{2} + \frac{1}{x}$.

*Proof.* This inequality is equivalent to $\ln(1+x) \geq \frac{2x}{x+2}$, or again $(x+2)\ln(1+x) \geq 2x$. Define $\phi : (-1, +\infty) \to \mathbb{R}$ as $\phi(x) = (x+2)\ln(1+x)$ and compute its derivatives:

$$\phi'(x) = \ln(1+x) + 1 + \frac{1}{1+x} \quad \text{and} \quad \phi''(x) = \frac{x}{(1+x)^2}.$$

We see that $\phi''(x) \geq 0$ for all $x \geq 0$, so $\phi$ is convex on $[0, +\infty)$. So we can use the tangent inequality:

$$\phi(x) \geq \phi(0) + \phi'(0)(x - 0) = 0 + 2x = 2x. \tag{24}$$

Therefore $\phi(x) \geq 2x$ for all $x \geq 0$, which is what we wanted to prove. □

### A.4    Rates for `SMGD` for Convex Lipschitz Functions: Proof of Proposition 5.4

**Proposition 5.4.** Let $\ell : \mathbb{R}^d \to \mathbb{R}$ be convex and $G$-Lipschitz, with argmin $\ell \neq \emptyset$. Let Assumptions 5.1 and 5.2 hold. Without loss of generality assume that $\delta_2 = \sigma_2 = 0$. Then, for $\eta = \frac{1}{(\delta_1 + \varepsilon)T + \sqrt{T}}$ and every $T \geq 1$, the iterates $\boldsymbol{w}_t$ from (16) satisfy

$$\mathbb{E}\left[\ell(\bar{\boldsymbol{w}}_T) - \inf \ell\right] = \mathcal{O}\left(\frac{1}{\sqrt{T}} + \delta_1 + \varepsilon\right),$$

where $\bar{\boldsymbol{w}}_T := \frac{1}{T}\sum_{t=0}^{T-1} \theta^t \boldsymbol{w}_t$ with $\theta = \frac{T}{T+2}$.

*Proof.* In this proof, we note $\mathcal{F}_t$ the filtration generated by $\boldsymbol{w}_0, \ldots, \boldsymbol{w}_t$, and we will note $\mathbb{E}_t$ to refer to the expectation conditioned on $\mathcal{F}_t$.

We want to apply Theorem A.4. Let us denote $\boldsymbol{g}_t = \boldsymbol{m}_t + \boldsymbol{e}_t$, so that the iterates of `SMGD` verify $\boldsymbol{w}_{t+1} = \boldsymbol{w}_t - \eta\boldsymbol{g}_t$. We are now going to show that $\boldsymbol{g}_t$ has uniformly bounded bias and variance. For this we will make use of Assumptions 5.1 and 5.2. Remember that we assumed $\delta_2 = \sigma_2 = 0$ without loss of generality (we can do this according to Lemma A.1), so we note $\delta$ and $\sigma$ instead of $\delta_1$ and $\sigma_1$. Also, Assumption 5.2 on the errors together with the tower rule imply that $\mathbb{E}_t\left[\|\boldsymbol{e}_t\|^2\right] = \mathbb{E}_t\left[\mathbb{E}\left[\|\boldsymbol{e}_t\|^2 \mid \boldsymbol{g}_t^{(1)}, \ldots, \boldsymbol{g}_t^{(n)}\right]\right] \leq \varepsilon^2$. Regarding the bias, we can write

$$\|\mathbb{E}_t\left[\boldsymbol{g}_t - \nabla\ell(\boldsymbol{w}_t)\right]\| \leq \|\mathbb{E}_t\left[\boldsymbol{m}_t - \nabla\ell(\boldsymbol{w}_t)\right]\| + \|\mathbb{E}_t\left[\boldsymbol{e}_t\right]\| \leq \delta + \varepsilon,$$

where in the last inequality we used Jensen's inequality to write $\|\mathbb{E}_t\left[e_t\right]\| \leq \sqrt{\mathbb{E}_t\left[\|e_t\|^2\right]}$. As for the variance, we write

$$\mathbb{E}_t\left[\|g_t - \mathbb{E}_t g_t\|^2\right] \leq 2\mathbb{E}_t\left[\|m_t - \mathbb{E}_t\left[m_t\right]\|^2\right] + 2\mathbb{E}_t\left[\|e_t - \mathbb{E}_t\left[e_t\right]\|^2\right] \leq 2\sigma^2 + 8\mathbb{E}_t\left[\|e_t\|^2\right] \leq 2\sigma^2 + 8\varepsilon^2.$$

Using Theorem A.4, we obtain

$$\mathbb{E}\left[\ell(\bar{w}_T) - \inf \ell\right] \leq \frac{2\|w_0 - w_*\|^2}{\eta T} + \eta\frac{2\sigma^2 + 8\varepsilon^2 + (\delta + \varepsilon + G)^2}{2} + \frac{\eta(\delta + \varepsilon)^2 T}{4}.$$

Setting the step size $\eta = \frac{1}{(\delta+\varepsilon)T+\sqrt{T}}$, we conclude that

$$\mathbb{E}\left[\ell(\bar{w}_T) - \inf \ell\right] \leq \frac{2D^2 + 2\sigma^2 + 8\varepsilon^2 + (\delta + \varepsilon)^2 + G^2}{\sqrt{T}} + (2D^2 + 1)(\delta + \varepsilon).$$

$\square$

### A.5 Rates for `SMGD` with Stable Noise: Proof of Corollary 5.6

**Corollary 5.6.** Let $n \geq 3$ be odd and $p = 1$. Consider the iterates from (16) with $e_t = 0$, and suppose that $g_t^{(i)} = \nabla\ell(w_t) + \Sigma(w_t)\zeta_t^{(i)}$ verifies Assumption 5.5.

(i) If $\ell$ is $L$-smooth and $\eta = \mathcal{O}(\frac{1}{\sqrt{T}})$, then we have that $\min_{0 \leq t \leq T-1} \mathbb{E}\left[\|\nabla\ell(w_t)\|^2\right] = \mathcal{O}(1/\sqrt{T})$.

(ii) If $\ell$ is convex and Lipschitz and $\eta = 1/\sqrt{T}$ then we have that $\mathbb{E}\left[\ell(\bar{w}_T) - \inf \ell\right] = \mathcal{O}(1/\sqrt{T})$, where $\bar{w}_T$ is defined as in Proposition 5.4.

*Proof.* First of all, remember that the $\ell_1$-median can be computed exactly, by computing component-wise a one-dimensional median. This allows us to have $\varepsilon = 0$ in Assumption 5.2. Further, due to Lemma A.3 we have $\delta_1 = 0$ in Assumption 5.1.

**Proof of part (i).** Choose $\eta = \frac{1}{8L(1+2\sigma_2+2\delta_2)\sqrt{T}}$, where $\delta_2, \sigma_2$ are from Lemma A.3. Hence, we can apply Proposition 5.3. As $\eta = \frac{C}{\sqrt{T}}$ with $C = \frac{1}{8L(1+2\sigma_2+2\delta_2)}$, we get the bound $\frac{1}{C\sqrt{T}} + \frac{C}{\sqrt{T}} = \mathcal{O}(1/\sqrt{T})$.

**Proof of part (ii).** The result is a direct consequence of Lemma A.3 and Proposition 5.4.

This completes the proof. $\square$

## B Approximate Computation of Gradient Estimators

### B.1 Gradient Estimation Problem: General Considerations

Given a function $\mathcal{D} : \mathbb{R}^d \to \mathbb{R}$ we consider the problem (5), which writes as

$$\underset{m \in \mathbb{R}^d}{\arg\min} \mathbb{E}\left[\mathcal{D}(m - z)\right].$$

In what follows, we will mostly consider functions $\mathcal{D}$ satisfying the following assumption.

**Assumption B.1.** The function $\mathcal{D} : \mathbb{R}^d \to \mathbb{R}$ is convex, $\underset{z}{\arg\min} \mathcal{D}(z) = \{0\}$ and problem (5) admits a solution.

Note that assuming $\mathcal{D}$ to be convex and finite implies that it is continuous, therefore measurable, which means that $\mathbb{E}\left[\mathcal{D}(m - z)\right]$ is well defined. The assumption that $\arg\min_z \mathcal{D}(z) = \{0\}$ guarantees that in the trivial case when the distribution is a Dirac $\delta_{m_0}$, then the solution to (5) is exactly $m_0$. The assumption that a solution to (5) exists is easily verified in practice, under the assumption that $\mathcal{D}$ is coercive:

**Lemma B.2.** Let $\mathcal{D} : \mathbb{R}^d \to \mathbb{R}$ be convex. Assume that $\mathbb{E}\left[\mathcal{D}(\boldsymbol{z} - \mathbb{E}\left[\boldsymbol{z}\right])\right] < +\infty$. If $\mathcal{D}$ is coercive, i.e.
$\lim\limits_{\|\boldsymbol{m}\|\to\infty} \mathcal{D}(\boldsymbol{m}) = +\infty$, then (5) admits a solution.

*Proof.* Let $\phi(\boldsymbol{m}) := \mathbb{E}\left[\mathcal{D}(\boldsymbol{m} - \boldsymbol{z})\right]$. Since we assume that $\phi(\mathbb{E}\left[\boldsymbol{z}\right]) = \mathbb{E}\left[\mathcal{D}(\boldsymbol{z} - \mathbb{E}\left[\boldsymbol{z}\right])\right] < +\infty$ then $\phi$ is proper. Since $\mathcal{D}$ is convex then $\phi$ is convex, as an expectation of convex functions. Moreover $\mathcal{D}$ is convex and finite, which means that $\mathcal{D}$ is continuous on $\mathbb{R}^d$, see e.g. Corollary 8.39 in Bauschke & Combettes (2017). Therefore $\phi$ is an expectation of lower semicontinuous functions, which implies that $\phi$ is lower semicontinuous.

Since $\mathcal{D}$ is coercive, convex and continuous, we know that there exists $a \in (0, +\infty)$ and $b \in \mathbb{R}$ (see Proposition 14.16 in Bauschke & Combettes (2017)) such that $\mathcal{D}(\cdot) \geq a\|\cdot\| + b$. Therefore

$$
\begin{aligned}
\phi(\boldsymbol{m}) &= \mathbb{E}\left[\mathcal{D}(\boldsymbol{m} - \boldsymbol{z})\right] \geq a\mathbb{E}\left[\|\boldsymbol{m} - \boldsymbol{z}\|\right] + b \\
&\geq a\|\boldsymbol{m} - \mathbb{E}\left[\boldsymbol{z}\right]\| - a\mathbb{E}\left[\|\boldsymbol{z} - \mathbb{E}\left[\boldsymbol{z}\right]\|\right] + b \\
&\geq a\|\boldsymbol{m} - \mathbb{E}\left[\boldsymbol{z}\right]\| - a\mathbb{E}\left[\mathcal{D}(\boldsymbol{z} - \mathbb{E}\left[\boldsymbol{z}\right])\right] \underset{\|\boldsymbol{m}\|\to\infty}{\longrightarrow} +\infty,
\end{aligned}
$$

where in the inequalities we used the fact that $\|\cdot\|$ is 1-Lipschitz, and that $a\mathbb{E}\left[\|\boldsymbol{z} - \mathbb{E}\left[\boldsymbol{z}\right]\|\right] \leq a\mathbb{E}\left[\mathcal{D}(\boldsymbol{z} - \mathbb{E}\left[\boldsymbol{z}\right])\right] - b < +\infty$. So $\phi$ is coercive on top of being proper convex and lower semicontinuous. We can then conclude that $\phi$ has a minimizer, with for instance Proposition 11.15 from Bauschke & Combettes (2017). $\square$

## B.2 Approximations Based on the Stochastic Proximal Point: General $\mathcal{D}$

Given a proper, closed convex function $\phi : \mathbb{R}^d \to \mathbb{R} \cup \{+\infty\}$ and $\tau > 0$, we recall the definition of the proximal operator

$$
\text{prox}_{\tau\phi}(\boldsymbol{m}_0) = \underset{\boldsymbol{m}\in\mathbb{R}^d}{\arg\min}\ \phi(\boldsymbol{m}) + \frac{1}{2\tau}\|\boldsymbol{m} - \boldsymbol{m}_0\|^2.
$$

Applying the SPP algorithm to (8) gives the following iterations

$$
\begin{cases}
\text{Sample } \boldsymbol{g}_t \text{ i.i.d. from } \mathcal{G} \\
\boldsymbol{m}_{t+1} = \underset{\boldsymbol{m}\in\mathbb{R}^d}{\arg\min}\ \mathcal{D}(\boldsymbol{m} - \boldsymbol{g}_t) + \frac{1}{2\tau}\|\boldsymbol{m} - \boldsymbol{m}_t\|^2.
\end{cases} \tag{25}
$$

As we see next, those iterations can be reformulated to simply involve the proximal operator of $\mathcal{D}$:

**Proposition B.3.** Let $\mathcal{D}$ verify Assumption B.1. The SPP update (25) for solving (9) can be equivalently written as

$$
\boldsymbol{m}_{t+1} = \boldsymbol{g}_t + \text{prox}_{\tau\mathcal{D}}(\boldsymbol{m}_t - \boldsymbol{g}_t) \tag{26}
$$

$$
= \text{prox}_{\tau\mathcal{D}^*(\frac{\boldsymbol{m}_t - \cdot}{\tau})}(\boldsymbol{g}_t), \tag{27}
$$

where $\mathcal{D}^*(\boldsymbol{m}) := \sup_{\boldsymbol{y}} \left(\langle \boldsymbol{y}, \boldsymbol{m} \rangle - \mathcal{D}(\boldsymbol{y})\right)$ is the Fenchel conjugate of $\mathcal{D}$, and $\boldsymbol{g}_t$ is sampled i.i.d. from $\mathcal{G}$ at each iteration.

*Proof.* Here we make use of proximal calculus rules, which can found in most textbooks on convex analysis, for example Chapter 6 in Beck (2017). Starting from (25), and applying the variable transformation $\boldsymbol{y} = \bar{\boldsymbol{y}} + \boldsymbol{g}_t$, we have

$$
\begin{aligned}
\boldsymbol{m}_{t+1} &= \underset{\boldsymbol{y}\in\mathbb{R}^d}{\arg\min}\ \mathcal{D}(\boldsymbol{y} - \boldsymbol{g}_t) + \frac{1}{2\tau}\|\boldsymbol{y} - \boldsymbol{m}_t\|^2 \\
&= \boldsymbol{g}_t + \underset{\bar{\boldsymbol{y}}\in\mathbb{R}^d}{\arg\min}\ \mathcal{D}(\bar{\boldsymbol{y}}) + \frac{1}{2\tau}\|\bar{\boldsymbol{y}} - (\boldsymbol{m}_t - \boldsymbol{g}_t)\|^2 \\
&= \boldsymbol{g}_t + \text{prox}_{\tau\mathcal{D}}(\boldsymbol{m}_t - \boldsymbol{g}_t).
\end{aligned}
$$

This proves (26). As for (27), we combine $(\tau\mathcal{D})^* = \tau\mathcal{D}^*(\cdot/\tau)$ (Beck, 2017, Thm. 4.14) and the Moreau decomposition theorem (Beck, 2017, Thm. 6.44), to obtain

$$\text{prox}_{\tau\mathcal{D}}(\boldsymbol{m}) = \boldsymbol{m} - \text{prox}_{\tau\mathcal{D}^*(\frac{\cdot}{\tau})}(\boldsymbol{m}).$$

Plugging into (26) we have

$$\begin{aligned}
\boldsymbol{m}_{t+1} &= \boldsymbol{g}_t + \text{prox}_{\tau\mathcal{D}}(\boldsymbol{m}_t - \boldsymbol{g}_t) \\
&= \boldsymbol{g}_t + \boldsymbol{m}_t - \boldsymbol{g}_t - \text{prox}_{\tau\mathcal{D}^*(\frac{\cdot}{\tau})}(\boldsymbol{m}_t - \boldsymbol{g}_t) \\
&= \boldsymbol{m}_t - \text{prox}_{\tau\mathcal{D}^*(\frac{\cdot}{\tau})}(\boldsymbol{m}_t - \boldsymbol{g}_t)
\end{aligned} \tag{28}$$

Finally, using that for any function $f$ it holds $\underset{\boldsymbol{y}}{\text{argmin}}\, f(\boldsymbol{y}) = \boldsymbol{m} + \underset{\bar{\boldsymbol{y}}}{\text{argmin}}\, f(\bar{\boldsymbol{y}} + \boldsymbol{m})$ in (28) gives

$$\begin{aligned}
\boldsymbol{m}_{t+1} &= \boldsymbol{m}_t - \left(\boldsymbol{m}_t + \text{prox}_{\tau\mathcal{D}^*(\frac{\cdot + \boldsymbol{m}_t}{\tau})}(-\boldsymbol{g}_t)\right) \\
&= -\text{prox}_{\tau\mathcal{D}^*(\frac{\cdot + \boldsymbol{m}_t}{\tau})}(-\boldsymbol{g}_t) \\
&= \text{prox}_{\tau\mathcal{D}^*(\frac{\boldsymbol{m}_t \cdot -}{\tau})}(\boldsymbol{g}_t)
\end{aligned}$$

where in the final equality we used that if $f(\boldsymbol{x}) = g(-\boldsymbol{x})$ we have that

$$\text{prox}_g(\boldsymbol{m}) = -\text{prox}_f(-\boldsymbol{m}).$$

$\square$

## B.3 Approximations Based on the Stochastic Proximal Point: Particular Cases of $\mathcal{D}$

We first characterize the SPP update (9) for when $\mathcal{D}$ is the $\ell_p$-norm, and show it simply require to project the sampled gradient onto a certain ball of radius $\tau$ centered at $\boldsymbol{m}_t$.

---

**Proposition B.4.** Let $\mathcal{D} = \|\cdot\|_p$ be the $\ell_p$-norm for $p \in [1, \infty]$. Let $q \in [1, \infty]$ such that $\frac{1}{p} + \frac{1}{q} = 1$, and let $\mathbb{B}_q(\boldsymbol{m}, \tau) := \{\boldsymbol{y} : \|\boldsymbol{m} - \boldsymbol{y}\|_q \leq \tau\}$. The SPP update (9) is given by

$$\boldsymbol{m}_{t+1} = \boldsymbol{m}_t + \text{Proj}_{\mathbb{B}_q(\boldsymbol{0}, \tau)}(\boldsymbol{g}_t - \boldsymbol{m}_t) \tag{29}$$

$$= \text{Proj}_{\mathbb{B}_q(\boldsymbol{m}_t, \tau)}(\boldsymbol{g}_t). \tag{30}$$

---

*Proof.* The proximal operator of the $\ell_p$-norm is given by

$$\text{prox}_{\tau\|\cdot\|_p}(\boldsymbol{m}) = \boldsymbol{m} - \tau\text{Proj}_{\mathbb{B}_q(\boldsymbol{0})}(\boldsymbol{m}/\tau),$$

see Example 6.47 in Beck (2017). Using this in (26) gives

$$\begin{aligned}
\boldsymbol{m}_{t+1} &= \boldsymbol{g}_t + \text{prox}_{\tau\|\cdot\|_p}(\boldsymbol{m}_t - \boldsymbol{g}_t) \\
&= \boldsymbol{m}_t - \tau\text{Proj}_{\mathbb{B}_q(\boldsymbol{0},1)}\left(\frac{\boldsymbol{m}_t - \boldsymbol{g}_t}{\tau}\right) \\
&= \boldsymbol{m}_t + \tau\text{Proj}_{\mathbb{B}_q(\boldsymbol{0},1)}\left(\frac{\boldsymbol{g}_t - \boldsymbol{m}_t}{\tau}\right) \\
&= \boldsymbol{m}_t + \text{Proj}_{\mathbb{B}_q(\boldsymbol{0},\tau)}(\boldsymbol{g}_t - \boldsymbol{m}_t) \\
&= \boldsymbol{m}_t + \left(\text{Proj}_{\mathbb{B}_q(\boldsymbol{m}_t,\tau)}(\boldsymbol{g}_t) - \boldsymbol{m}_t\right) \\
&= \text{Proj}_{\mathbb{B}_q(\boldsymbol{m}_t,\tau)}(\boldsymbol{g}_t)
\end{aligned}$$

where the third equality we used that

$$\text{Proj}_{\mathbb{B}_q(\boldsymbol{0},1)(\boldsymbol{m})} = -\text{Proj}_{\mathbb{B}_q(\boldsymbol{0},1)(-\boldsymbol{m})},$$

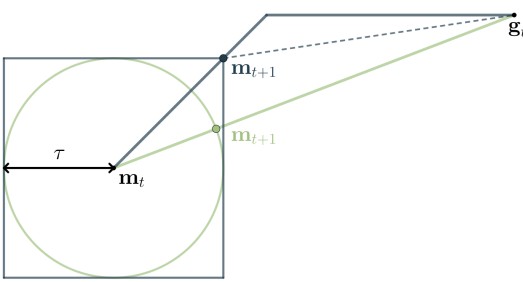

Figure 7: One $\mathtt{SPP}$ update (30) for $\mathcal{D} = \|\cdot\|_1$ (blue) or $\mathcal{D} = \|\cdot\|_2$ (green) amounts to project onto a ball of the dual norm, centered at the current iterate $\boldsymbol{m}_t$ and of radius $\tau > 0$. The thick lines represent the projections of $\boldsymbol{g}_t$ for all possible values of $\tau$.

followed by

$$\mathrm{Proj}_{\mathbb{B}_q(\boldsymbol{m}/\tau,1)}(\boldsymbol{y}/\tau) = \frac{1}{\tau}\mathrm{Proj}_{\mathbb{B}_q(\boldsymbol{m},\tau)}(\boldsymbol{y})$$

in the fourth equality and

$$\mathrm{Proj}_{\mathbb{B}_q(\boldsymbol{0},\tau)}(\boldsymbol{y}) = \mathrm{Proj}_{\mathbb{B}_q(\boldsymbol{m},\tau)}(\boldsymbol{y} - \boldsymbol{m}) - \boldsymbol{m}$$

in the fifth equality. $\qquad\square$

Using Proposition B.4 we can now develop closed form updates of the $\mathtt{SPP}$ method for when $\mathcal{D}$ is the $\ell_1$- or the $\ell_2$-norm, with both methods being related to clipping.

**Corollary 3.2.** For $\mathcal{D} = \|\cdot\|_1$ update (9) is given by

$$\boldsymbol{m}_{t+1} = \boldsymbol{m}_t + \mathrm{clip}_{\tau,1}(\boldsymbol{g}_t - \boldsymbol{m}_t), \tag{11}$$

where $\mathrm{clip}_{\tau,1}(\boldsymbol{v}) := (\min\{\max\{\boldsymbol{v}_i, -\tau\}, \tau\})_{i=1}^d$.

*Proof.* Follows from (29) and that

$$\min\{\max\{\boldsymbol{v}, -\tau\}, \tau\} = \mathrm{Proj}_{\mathbb{B}_\infty(\boldsymbol{0},\tau)}(\boldsymbol{v}).$$

$\qquad\square$

**Corollary 3.1.** For $\mathcal{D} = \|\cdot\|_2$ update (9) is given by

$$\boldsymbol{m}_{t+1} = \boldsymbol{m}_t + \mathrm{clip}_{\tau,2}(\boldsymbol{g}_t - \boldsymbol{m}_t), \tag{10}$$

where $\mathrm{clip}_{\tau,2}(\boldsymbol{v}) := \frac{\tau}{\max\{\tau, \|\boldsymbol{v}\|_2\}}\boldsymbol{v}$.

*Proof.* Follows from (29), by using that

$$\mathrm{Proj}_{\mathbb{B}_2(\boldsymbol{0},\tau)}(\boldsymbol{v}) = \frac{\tau\,\boldsymbol{v}}{\max\{\tau, \|\boldsymbol{v}\|_2\}}.$$

$\qquad\square$

We end this section with the closed form updates of the $\mathtt{SPP}$ method for when $\mathcal{D}$ is the Huber function.

**Corollary 3.3.** For $\mathcal{D} = H_\mu$, update (9) is given by

$$\boldsymbol{m}_{t+1} = \boldsymbol{g}_t + \beta_t(\boldsymbol{m}_t - \boldsymbol{g}_t) = \beta_t\boldsymbol{m}_t + (1 - \beta_t)\boldsymbol{g}_t, \tag{12}$$

where $\beta_t := 1 - \frac{\mu\tau}{\max\{\|\boldsymbol{m}_t - \boldsymbol{g}_t\|,\ \mu(1+\tau)\}}$.

*Proof.* Follows from (9) and the fact that the proximal operator of the Huber function is given by $\mathrm{prox}_{\tau H_\mu}(\boldsymbol{z}) = \left(1 - \frac{\mu\tau}{\max\{\|\boldsymbol{z}\|, \mu(1+\tau)\}}\right)\boldsymbol{z}$. For a proof, see Beck (2017, Example 6.66), and note that here we have the additional factor $\mu$ in the definition of $H_\mu$. □

## B.4   Approximations Based on Subgradient Updates

Instead of SPP, we could solve (8) by iteratively taking steps of stochastic subgradient descent. If $\mathcal{D}$ is convex, then in each iteration we sample a gradient $\boldsymbol{g}_t$, and update our current estimate $\boldsymbol{m}_t$ by

$$\boldsymbol{m}_{t+1} = \boldsymbol{m}_t - \tau\boldsymbol{u}_t, \quad \boldsymbol{u}_t \in \partial\mathcal{D}(\boldsymbol{m}_t - \boldsymbol{g}_t). \tag{31}$$

Here, $\tau > 0$ is the learning rate, and $\boldsymbol{u}_t$ is a subgradient (of the convex subdifferential). Since $\mathcal{D}$ could be a non-differentiable function such as the $\ell_2$- or $\ell_1$-norm, we use subgradients instead of gradients.

**Squared $\ell_2$-norm.**   As a first simple example of (31), let $\mathcal{D} = \frac{1}{2}\|\cdot\|_2^2$. In this case (31) becomes

$$\boldsymbol{m}_{t+1} = \boldsymbol{m}_t - \tau(\boldsymbol{m}_t - \boldsymbol{g}_t) = (1-\tau)\boldsymbol{m}_t + \tau\boldsymbol{g}_t. \tag{32}$$

This is again (heavy-ball) momentum (cf. SGD-M), now with coefficient $\beta = 1 - \tau$. Note that in contrast to SPP, the coefficient $\beta$ could be negative, and is only in $[0,1)$ (as typically is the case for momentum) if $\tau \in [0,1]$.

**$\ell_1$-norm.**   If we choose $\mathcal{D} = \|\cdot\|_1$, update (31) gives

$$\boldsymbol{m}_{t+1} = \boldsymbol{m}_t + \tau\,\mathrm{sgn}(\boldsymbol{g}_t - \boldsymbol{m}_t),$$

where the sgn-operator can take any value in $[-1,1]$ for coordinates $i \in [d]$ such that $(\boldsymbol{g}_t - \boldsymbol{m}_t)_i = 0$. This recovers a version of sign-SGD (Riedmiller & Braun, 1993; Bernstein et al., 2018), where the sign-operation is applied to the increment $\boldsymbol{m}_t - \boldsymbol{g}_t$. For example, if we reset $\boldsymbol{m}_t = 0$ in every iteration, we obtain exactly sign-SGD.

**$\ell_2$-norm.**   If we choose $\mathcal{D} = \|\cdot\|_2$, and if $\boldsymbol{g}_t \neq \boldsymbol{m}_t$, update (31) gives

$$\boldsymbol{m}_{t+1} = \boldsymbol{m}_t - \tau\frac{\boldsymbol{m}_t - \boldsymbol{g}_t}{\|\boldsymbol{m}_t - \boldsymbol{g}_t\|} = \left(1 - \frac{\tau}{\|\boldsymbol{m}_t - \boldsymbol{g}_t\|}\right)\boldsymbol{m}_t + \frac{\tau}{\|\boldsymbol{m}_t - \boldsymbol{g}_t\|}\boldsymbol{g}_t.$$

Otherwise, if $\boldsymbol{g}_t = \boldsymbol{m}_t$, we can choose any $\boldsymbol{u}_t$ with $\|\boldsymbol{u}_t\| \leq 1$ and set $\boldsymbol{m}_{t+1} = \boldsymbol{m}_t - \tau\boldsymbol{u}_t$; in particular we can set $\boldsymbol{m}_{t+1} = \boldsymbol{m}_t$.

## B.5   Rates for SPP

Now that we have presented how to compute the iterates (9) in practice, let us state its convergence properties. Since all these methods are instantiates of the SPP method, we can use standard results such as Davis & Drusvyatskiy (2019, Thm. 4.4).

**Proposition B.5.** Let $\mathcal{D}$ be a norm, let $\phi(\boldsymbol{m}) := \mathbb{E}_{\boldsymbol{g}\sim\mathcal{G}}[\mathcal{D}(\boldsymbol{m} - \boldsymbol{g})]$, and let $\boldsymbol{m}_* \in \underset{\boldsymbol{m}\in\mathbb{R}^d}{\mathrm{argmin}}\ \phi(\boldsymbol{m})$. If $\boldsymbol{m}_t$ is generated by (9) then for $\bar{\boldsymbol{m}}_T = \frac{1}{T}\sum_{t=1}^T \boldsymbol{m}_t$ we have

$$\mathbb{E}[\phi(\bar{\boldsymbol{m}}_T) - \phi(\boldsymbol{m}_*)] = O\left(\frac{\|\boldsymbol{m}_0 - \boldsymbol{m}_*\|^2}{\tau T} + \tau\right).$$

*Proof.* Since we assume that $\mathcal{D}$ is a norm over $\mathbb{R}^d$ then it convex and continuous. Moreover, since all norms are equivalent in $\mathbb{R}^d$, we have that $\mathcal{D}(\boldsymbol{w}) \leq C\|\boldsymbol{w}\|_2$ for some constant $G > 0$. Thus $\mathcal{D}$ is a $G$-Lipschitz function. If now we write $\phi_{\boldsymbol{g}} := \mathcal{D}(\cdot - \boldsymbol{g})$, it is clear that it is convex and Lipschitz. Therefore, $\phi = \mathbb{E}_{\boldsymbol{g}}[\phi_{\boldsymbol{g}}]$ is itself a convex $G$-Lipschitz function. The claim now follows from Davis & Drusvyatskiy (2019, Theorem 4.4). □

Here is a corollary stating rates on the expected distance between the approximate sample median and the true sample median, when using the $\ell_1$-norm:

> **Corollary B.6.** In the context of Proposition B.5, assume that $\mathcal{D} = \|\cdot\|_1$ and that $\mathcal{G}$ is a sum of Dirac measures $\frac{1}{n}\sum_{i=1}^n \delta_{\boldsymbol{g}^{(i)}}$. Then we have further
>
> $$\mathbb{E}\left[\|\bar{\boldsymbol{m}}_T - \boldsymbol{m}_*\|\right] = O\left(\frac{\|\boldsymbol{m}_0 - \boldsymbol{m}_*\|^2}{\tau T} + \tau\right).$$

*Proof.* In this proof, we use the notation $(\boldsymbol{x}_j)_{j=1,\ldots,d}$ to specify the indivdual components of a vector $\boldsymbol{x} \in \mathbb{R}^d$. For $\boldsymbol{x} \in \mathbb{R}^d$, we define the sign operator as follows: let $\operatorname{sgn}(\boldsymbol{x}) = (\operatorname{sgn}(\boldsymbol{x}_j))_{j=1,\ldots,d}$, where $\boldsymbol{x}_j$ is the $j$-th coordinate of $\boldsymbol{x} \in \mathbb{R}^d$, and where we define for a scalar $t \in \mathbb{R}$ the set-valued operator $\operatorname{sgn}(t) = \{+1\}$ if $t > 0$, $\operatorname{sgn}(t) = \{-1\}$ if $t < 0$, and $\operatorname{sgn}(t) = [-1, 1]$ if $t = 0$. Further, given a set $A \subset \mathbb{R}^d$ we define $\|A\|_- := \inf_{a \in A}\|a\|$.

To obtain the desired bound, and given that we already have the conclusion of Proposition B.5, it is enough for us to show that there exists $\mu > 0$ such that

$$(\forall \boldsymbol{m} \in \mathbb{R}^d) \quad \mu \operatorname{dist}(\boldsymbol{m}, \operatorname{argmin} \phi) \leq \phi(\boldsymbol{m}) - \inf \phi.$$

It is a standard result from variational analysis that this property[6] is true if and only if the following inequality is satisfied (see Theorem 5.1 in Cornejo et al. (1997) or Proposition 3.1 in Lemaire (1998))

$$(\forall \boldsymbol{m} \in \operatorname{dom} \partial\phi) \quad \|\partial\phi(\boldsymbol{m})\|_- \geq \mu.$$

Our goal now is to prove the above inequality, from which the conclusion will follow. Let us consider $\boldsymbol{m} \notin \operatorname{argmin} \phi$, and use standard calculus to write

$$\partial\phi(\boldsymbol{m}) = \frac{1}{n}\sum_{i=1}^n \operatorname{sgn}(\boldsymbol{m} - \boldsymbol{g}_i) = \frac{1}{n}\sum_{i=1}^n \left(\operatorname{sgn}(\boldsymbol{m}_j - (\boldsymbol{g}_i)_j)\right)_{j=1,\ldots,d} = \frac{1}{n}\left(\sum_{i=1}^n \operatorname{sgn}(\boldsymbol{m}_j - (\boldsymbol{g}_i)_j)\right)_{j=1,\ldots,d}.$$

Our goal is to show that $\|\partial\phi(\boldsymbol{m})\|_- \geq \frac{1}{n}$, which is equivalent to show that $\|\sum_{i=1}^n \operatorname{sgn}(\boldsymbol{m} - \boldsymbol{g}_i)\|_- \geq 1$. Because $\boldsymbol{m} \notin \operatorname{argmin} \phi$, we know that $0 \notin \partial\phi(\boldsymbol{m})$ so there must exist a coordinate $j$ such that the $j$-th coordinate of $\partial\phi(\boldsymbol{m})$ does not contain zero, in other words such that $0 \notin \sum_{i=1}^n \operatorname{sgn}(\boldsymbol{m}_j - (\boldsymbol{g}_i)_j)$. Using the fact that $\|\cdot\|_2 \geq \|\cdot\|_\infty$, we can therefore lower bound $\|\sum_{i=1}^n \operatorname{sgn}(\boldsymbol{m} - \boldsymbol{g}_i)\|_- \geq |\sum_{i=1}^n \operatorname{sgn}(\boldsymbol{m}_j - (\boldsymbol{g}_i)_j)|_-$. Let us denote $s_i := \operatorname{sgn}(\boldsymbol{m}_j - (\boldsymbol{g}_i)_j)$, so that we want to prove that $|\sum_{i=1}^n s_i|_- \geq 1$. Let us now consider a few cases.

- If $s_i = \{+1\}$ or $\{-1\}$ for every $i$, then $\sum_{i=1}^n s_i$ is a singleton. Furthermore it is a sum of relative numbers, so $\sum_{i=1}^n s_i \in \mathbb{Z}$. But we also know that $0 \notin \sum_{i=1}^n s_i$, so we conclude that $\sum_{i=1}^n s_i \in \mathbb{Z}^*$, and so that $|\sum_{i=1}^n s_i|_- \geq 1$.

- If $s_i = [-1, +1]$ for every $i$, then we immediately see that $0 \in \sum_{i=1}^n s_i$ which is a contradiction.

- Otherwise, the $s_i$ are combinations of singletons and intervals. Let us note $I = \{i : s_i = [-1, +1]\}$ and $I' = \{i : s_i = \{\pm 1\}\}$, which are not empty by assumption. Let us also note $k \geq 1$ the cardinality

---

[6]It is sometimes referred to as (weak) sharp minima (Ferris, 1991; Burke & Ferris, 1993), (superlinear)(linear) conditioning (Lemaire, 1992; Cornejo et al., 1997; Lemaire, 1998), or error bound (Lewis & Pang, 1998).

of $I$, so that $\sum_{i \in I} s_i = [-k, +k]$. We also introduce $s := \sum_{i \in I'} s_i \in \mathbb{Z}$. Now we can write

$$|\sum_{i=1}^{n} s_i|_{-} = \inf_{t \in \sum_{i=1}^{n} s_i} |t| = \inf_{t \in [-k,+k]} |t + s| = d(s; [-k, +k]),$$

where the latter is the distance from the number $s$ to the interval $[-k, +k]$. Now remember that $0 \notin \sum_{i=1}^{n} s_i$, so this distance cannot be zero, which means that $|s| > k$. Moreover both $s$ and $k$ are rational numbers, so this distance must be greater or equal to 1. This concludes the proof.

$\square$

## C   Supplementary Information on Experiments

This section provides additional information for the experimental setup.

For the language modeling experiments, all details are identical to Kunstner et al. (2023), Section A.1. They also provide implementations for all tasks at https://github.com/fKunstner/noise-sgd-adam-sign, which we use. For the training runs for language modeling, we use batch size 256 for PTB, 320 for WikiText-2, and 32 for SQuAD.

### C.1   Additional Plots

**Ablations for Fig. 3**   . We provide two ablation studies for the experiment on gradient estimation on heavy-tailed synthetic data. First, Fig. 8 (left) shows that for Gaussian noise ($\alpha = 2$), the performance of momentum can surpass V/CClip depending on selection of the step size $\tau$. However, this does not fix the issue of a quickly degrading performance for momentum, when $\alpha$ decreases.

Second, Fig. 8 (right) shows the performance under *skewed* heavy-tailed noise; for this, we set the skewness parameter of the $\alpha$-stable distribution to one.

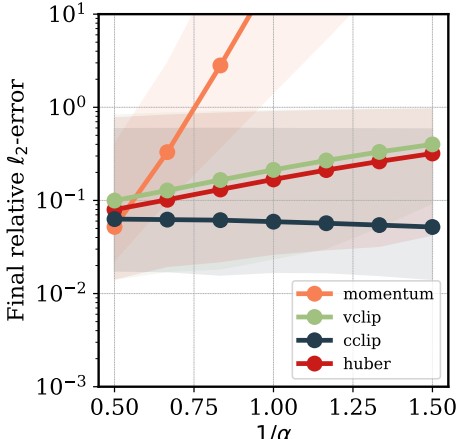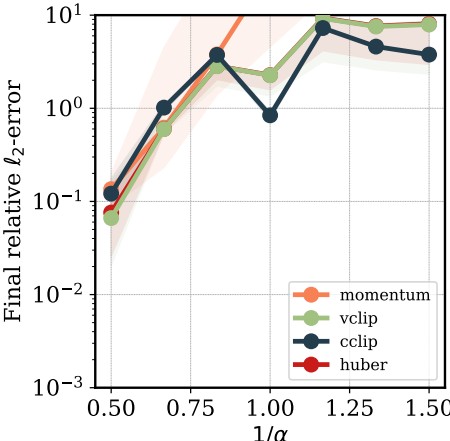

Figure 8: **(Left)** Same as Fig. 3 but with $\tau = 0.001$. Here, momentum performs best for Gaussian data, but still becomes instable when the noise is more heavy-tailed. **(Right)** Same as Fig. 3 but with *skewed* noise. Here, the median of the noise is no longer zero, which explains the failure of V/CClip.

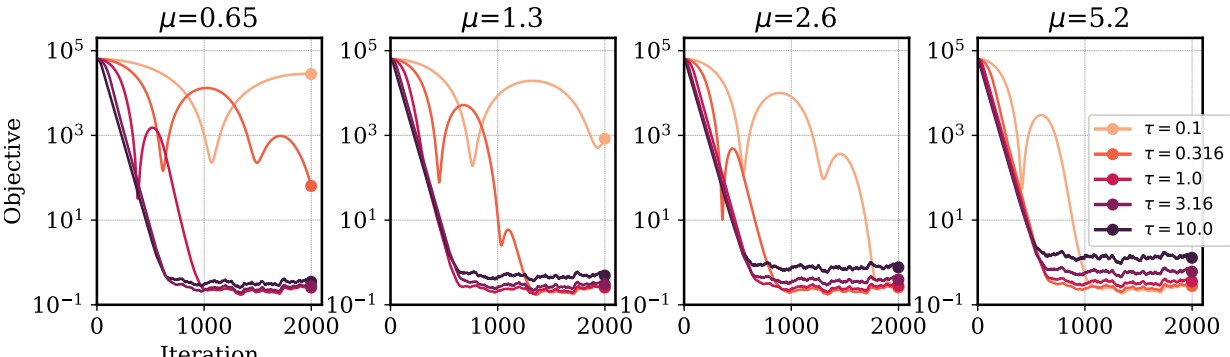

Figure 9: Least-squares problem (same as (S1)) for `Huber` with different values of $\mu$ and $\tau$. Setting $\tau = 1.0$ performs relatively well across the selected range of values of $\mu$.

**Full gradient estimation with fixed weights.** Here, we show the results for estimating the full gradient of a transformer architecture, with weights fixed at initialization. Fig. 10 shows the results for `PTB` where we tried three different batch sizes $\{64, 256, 1024\}$. Fig. 11 shows the results for `WikiText-2`.

We remark that in this experiment only, we have used the first order approximation of (7), namely $\boldsymbol{m}_{t+1} = (1 - \tau)\,\boldsymbol{m}_t + \tau\boldsymbol{g}_t$. That is, we replace $\frac{\tau}{1+\tau}$ with its first-order Taylor approximation around zero, which is equal to $\tau$. As the value of $\tau$ is small, this has negligible impact on the result: for example, if $\tau = 0.01$, then $\frac{\tau}{1+\tau} \approx 0.0099$.

*Discussion.* From Fig. 10, we observe two phenomena: in the long run, momentum attains the lowest error for estimating the full batch gradient. However, the initial decrease of the error is much faster for `CClip`, followed by `VClip`. This is important when using these estimates within a training setup such as (13), where we only do one iteration of gradient estimation, followed by an update of the weight (and hence a change in the full-batch gradient). Secondly, we observe that the difference in convergence speeds is most pronounced when the batch size increases. Hence, being robust to outliers seems not to be solvable only by increasing the batch size and thus decreasing the noise of the mini-batch gradient. In fact, the contrary seems to be the case. This is similar to the observations made in Kunstner et al. (2023).

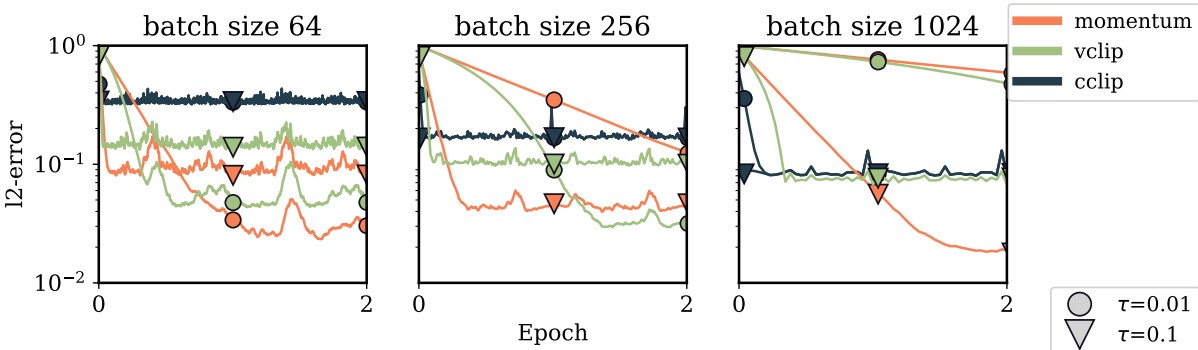

Figure 10: Encoder transformer on `PTB` dataset, with weights fixed at initialization. We use momentum, $\text{clip}_{\tau,1}$ and $\text{clip}_{\tau,2}$ to estimate the full-batch gradient.

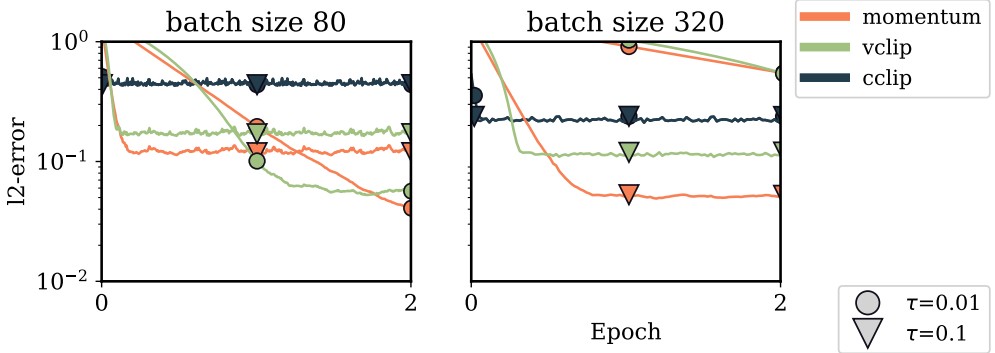

Figure 11: Encoder transformer on `Wikitext-2` dataset, with weights fixed at initialization. We use momentum, $\text{clip}_{\tau,1}$ and $\text{clip}_{\tau,2}$ to estimate the full-batch gradient.

## C.2 Learning Rate Values for Transformer Training

Here, we report for each of the language modelling tasks, the learning rate value that is displayed in Fig. 5. For each method, we tuned the learning rate over a set of values $10^{\frac{j}{2}}$ for a suitable interval of integer numbers $j \in \mathbb{Z}$.

Table 1: Tuned learning rate values used for each method. All methods use constant learning rates. For `SGD-M`, tuning information is also reported in Section C.1 in (Kunstner et al., 2023)).

| Name | SGD-M | clipped-SGD | VClip | CClip | Adam |
|---|---|---|---|---|---|
| PTB | 1.0 | 3.16 | 3.16 | 0.316 | 0.001 |
| WikiText-2 | 0.316 | 3.16 | 1.0 | 0.316 | 0.001 |
| SQuAD | 0.316 | 1.0 | 0.316 | 0.1 | 0.000316 |

