# OpenReview forum: "Tracking the Median of Gradients with a Stochastic Proximal Point Method"
_TMLR — Accepted by TMLR_

### Review · Reviewer_oT8M · 2025-08-05

**Summary Of Contributions:**

This paper studies convex optimization under noisy gradient feedback, focusing specifically on methods for *robustly estimating gradients* when the noise is unfavorable—such as when it contains significant outliers, exhibits heavy tails, or is adversarially corrupted. Motivated by the median trick in robust statistics, the authors propose a general framework based on the *stochastic proximal point (SPP)* method, using different proximity functions (e.g., $\ell_1$, $\ell_2$, or squared $\ell_2$ norms). The paper shows that several classical methods—such as momentum, gradient clipping, and Huber-style updates—can all be recovered as special cases of this SPP framework by choosing appropriate prox operators.

Under certain assumptions, the authors prove that if an oracle can provide a few independent gradient samples per iteration, then the proposed method (using sample *median* estimate for gradients) achieves the standard $O(1/\sqrt{T})$ convergence rate—even under $\alpha$-stable or state-dependent noise, where variance may be infinite and standard SGD can diverge.

Synthetic experiments are conducted to validate the robustness of these estimators under heavy-tailed noise. The results show that median-based updates are more stable and accurate than momentum or clipped-SGD in pathological regimes. Additional experiments on real NLP tasks (e.g., PTB, WikiText-2, and SQuAD) show modest improvements over momentum, although the method does not outperform Adam.

**Audience:**

Yes

**Claims And Evidence:**

Yes

**Requested Changes:**

I outlined some minor comments in the weaknesses section, but they do not affect my overall recommendation. I believe this is a solid work that makes a clear conceptual and theoretical contribution. It is suitable for publication in TMLR.

**Strengths And Weaknesses:**

**Strengths:**

1. The writing is very clean and easy to follow.
2. The framework based on the stochastic proximal point method for estimating gradients appears to be a novel contribution that unifies several existing approaches.
3. The paper provides both theoretical analysis and empirical validation. The convergence guarantees under heavy-tailed and state-dependent noise are clearly stated and backed by synthetic and real-world experiments.

**Weaknesses:**

1. The technical depth does not appear to be very high; the convergence analysis seems fairly standard. That said, I am only marginally familiar with the relevant literature, so I have limited expertise to comment definitively on its originality or significance.
2. The empirical gains on real tasks are modest. While the proposed methods outperform momentum in some settings, they do not surpass Adam, and the robustness benefits are mostly visible in synthetic or controlled setups.
3. The paper could benefit from more discussion of practical guidelines—for example, how to choose appropriate proximity functions for different contexts, or when the proposed methods are likely to outperform existing optimizers in real-world applications.

---

> ### Author Response · Authors · 2025-09-18
> **Author Response**
>
> Dear reviewer,
>
> Thank you for the feedback and questions on our submission. We are happy to hear that the submission is considered adequate for publication.
>
>
> > the convergence analysis seems fairly standard
>
> The reviewer is correct in that the proof techniques of Proposition 5.3 and 5.4 indeed are similar to standard techniques in proofs for biased SGD. The focus and novelty of our analysis mainly circles around situations where bias and variance assumptions can be satisfied under certain heavy-tailed noise distributions for the median, but not for the mean. One concrete realization of this is Corollary 5.6, which allows state-dependent heavy-tailed noise for the sample median method; we are not aware of other convergence results for such a general setting.
>
> In any case, we will state more clearly in the main text that our proofs follow standard techniques from analysing biased SGD, instead of only mentioning it in the proof sections (see the beginning of Section A.3).
>
> > The empirical gains on real tasks are modest, do not surpass Adam
>
> We agree with the reviewer on this point. Despite this, we think that a valuable contribution of the paper is to reveal the connection between existing methods and their relation to mean/median estimation; further, the methods we present can be useful in applications outside of language modelling, which we haven’t tested or are not aware of.
>
>
> > how to choose appropriate proximity functions for different contexts
>
> Being able to identify the correct proximity function for a given task is not trivial and usually requires domain knowledge. For example, many papers in signal/image processing revolve around finding the proper proximity function modeling a given problem (dealing with a particular kind of noise, or inducing a particular prior on some feature). We think that such recommendations would be out of scope for our paper, which focuses on standard proximity functions and connects them with existing algorithm tricks such as clipping.

---

### Review · Reviewer_5XN1 · 2025-08-27

**Summary Of Contributions:**

The authors of this paper study the optimization algorithms based on the estimated median of stochastic gradients, whose convergence is more robust to heavy-tailed gradient noises.

1. They study the mean/median gradient estimation problems using stochastic proximal point (SPP) methods under a fixed gradient distribution. As a result, they figure out that some notions of gradient clipping are related to estimating the median of gradients online.
2. Alternately applying a SPP update (of mean/median estimates) and a descent step (to update the learning parameters), they recover several existing clipping-based algorithms (especially for distributed learning scenarios) as special cases.
3. Pivoting the interest to the sample median gradient descent (SMGD) algorithm, they show that the convergence of SMGD is guaranteed more robustly even under heavy-tailed, state-dependent gradient noise.

Lastly, they numerically compare the algorithms based on mean/median gradient estimations on toy examples and language modeling tasks.

**Audience:**

Yes

**Broader Impact Concerns:**

Since this work mostly focuses on the theoretical aspects of optimization algorithms and their application in toy problems, the Broader Impact Statement section seems unnecessary.

**Claims And Evidence:**

No

**Requested Changes:**

**Required Changes**

- Every point listed above under “weaknesses” should be resolved.

**Recommended Changes & Some Questions expected to be answered**

1. The paper is mainly comparing $\mathcal{D}=\lVert\cdot\rVert^2$ versus $\mathcal{D}=\lVert\cdot\rVert$. Can you provide some discussions about what happens if we let $\mathcal{D}=\lVert\cdot\rVert^\gamma$ where $1<\gamma<2$ or even $0 < \gamma < 1$?
2. Questions regarding Figure 3
    1. Why does the “momentum” have a worse $\ell_2$ error than the other clipping-based method even when $1/\alpha=0.5$? As far as I know, the standard $\alpha$-stable distribution reduces to a Gaussian distribution when $\alpha=2$, which is light-tailed (having a finite variance). I do not see any reason for the mean-estimating method being worse than the median-estimating method(s), even under the light-tailed noise.
    2. What will happen in the plots if we encounter a *skewed* (but heavy-tailed) noise distribution, having the median different from the mean?
3. Questions regarding Figure 4
    1. The method “huber” must trade off between mean-/median-estimating methods. Have you tried to draw the objective plots by manipulating the value of the parameter $\mu$ for the Huber function?
    2. Will a smaller learning rate or a learning rate scheduling alleviate the instability in SGD with sample mean gradients?
4. Comments regarding Figures 5 and 6
    1. Adam with a usual gradient clipping, VClip, or CClip can be put into the baselines. In particular, we often use Adam with gradient clipping in real-world LLM training.
    2. Probably, $\ell_1$-SMGD can also be tested for training language models. ($\ell_2$-SMGD is not that necessary due to its computational burden.)
5. Minor typo under Figure 6: “$\texttt{V/CCLip}$” $\rightarrow$ ”$\texttt{V/CClip}$”.

**Strengths And Weaknesses:**

**Strengths**

1. The paper is well-written and easy to follow. It provides a set of nice preliminaries about $\ell_p$-medians of random vectors and their online estimation problems. I find the relationship between gradient clipping and online median gradient estimation problems very interesting.
2. This paper proposes an insightful general viewpoint for analyzing several existing clipping-based optimization algorithms under the framework of online median gradient estimation.
3. The assumptions for the theoretical analysis in Section 5 and the discussions therein are clear and sound.
4. This paper provides a set of systematic comparative analyses between several algorithms based on (or, at least, related to) mean/median gradient estimation, which clearly shows that median-based and clipping-based algorithms showcase a robust optimization dynamic in general.

**Weaknesses**

1. **The relationship between the online median gradient estimation and the robustness of several clipping-based algorithms (e.g., V/CClip-SGD) is not very convincing.**
    1. Throughout the paper, the authors emphasize the benefits of clipping-based methods in terms of robust convergence, attributing it to the online median estimation of the stochastic gradients. As kindly explained in Section 3, the update rule of $\boldsymbol{m}_t$ based on vector-/component-wise clipping leads to the estimated median gradient only when the distribution $\mathcal{G}$ of the gradients is fixed. On the other hand, if the update rule is “interweaved” with the weight update rule, it will provide an estimate of the median gradient over the course of the training trajectory. This is not quite the same as the median of the stochastically sampled (possible) update directions at each time step. Also, we do not really want to estimate the median of all the stochastic gradients that (can) appear during training because (1) the distribution of stochastic gradients must change depending on the weight vector $\boldsymbol{w}_t$; (2) we expect the gradients (or the amount/direction of the update) to be very small at near convergence, mostly at the end of the training. In this sense, I do not believe that the vector-/component-wise clipping (as in the paper) will provide robustness in the training trajectory due to estimating the median gradients (yet there might be some other reasons).
2. **Section 5 does not provide genuine convergence guarantees for SMGD.**
    1. As far as I understand, the proofs of Propositions 5.3 and 5.4 do not rely on the fact that $\boldsymbol{m}_t$ is the sample $\ell_p$-**median** of stochastic gradients. The proofs always hold when Assumptions 5.1 and 5.2 hold, where we can replace the term “sample median” with any kind of random vector or any estimator (in a statistical sense) of the loss gradient $\nabla \ell (\boldsymbol{w})$. The paper lacks such an explanation that the propositions are generally true for any kind of stochastic gradient method with biased gradient estimators.
    1. According to the discussion in the main text, the authors claim that it is important to use the estimator of median gradients because it generally has smaller (finite) variance given that the sample size $n$ is sufficiently large. This claim is intuitively correct, but the paper does not handle this topic rigorously. Several questions remain: Exactly when (e.g., for what class of distributions, or with how large a sample size) does the variance of the estimator become reasonably small in general? Do we have a theoretical/rigorous guarantee that the methods using the sample mean gradients always fail to converge under heavy-tailed (or state-dependent) gradient noise? Is it always true that SGD using the sample median gradients succeeds in converging when SGD using the sample mean gradients can succeed?
    1. Propositions 5.3 and 5.4 have irreducible terms in $\delta_1$ and $\varepsilon$, each due to the bias of the gradient estimator and the error for estimating the sample median. These terms can be huge in general, meaning that the propositions do not really prove a meaningful convergence to a (local) optimum. Even though Corollary 5.6. seems to provide a couple of convergence results using a step size of scale $1/\sqrt{T}$,  I strongly believe that this is possible because of Assumption 5.5, supposing an **unbiased** gradient oracle (although its distribution is heavy-tailed). In general, the mean and the median of a distribution do not match at all. I believe analyzing the behavior of the optimization algorithms must be significant, especially when the median is quite (or even largely) apart from the mean, and in this case, I think we need convergence metrics other than “mean” squared gradient norm or “mean” sub-optimality in function values.
        - By the way, the step size conditions in Propositions 5.3 and 5.4 are not reflected in Corollary 5.6.
    1. Lastly, the descriptions of Propositions 5.3 and 5.4 lack clear dependencies regarding the sample size $n$ (of course, the proofs of these propositions do not depend on the sample size $n$ in general) and a problem parameter $\sigma_1$.
3. **Algorithmic behaviors of V/CClip-SGD are not studied well.**
    1. Although the relationship between vector-/component-wise clipping gradients under fixed gradient distribution and the online median gradient estimation is interesting, a more interesting topic would be the behavior of optimization algorithms when applying these clipping techniques. Unfortunately, this aspect is not well studied in this paper.
    1. There can be several research questions. Is the median estimation really happening when we interweave the clipping-based SPP update and the weight update? How does the estimated ‘median’ ($\boldsymbol{m}_t$) behave? How good/bad/similar are they compared to SGD using the sample mean/median gradients?
    1. The only thing I can expect from Propositions 5.3 and 5.4 is that V/CClip-SGD are *strictly* worse than SGD with exact sample median gradients ($\varepsilon = 0$) due to the estimation error for the sample median.

---

> ### Author Response · Authors · 2025-09-18
> **Author response**
>
> Dear reviewer,
>
> Thank you for the detailed feedback on our paper. We respect to each point/question below:
>
> > we expect the gradients (or the amount/direction of the update) to be very small at near convergence
>
> Here we disagree: in the stochastic setting, and unless interpolation holds, the batch gradients will not be small, which is practically the case for many machine learning problems.
>
> > In this sense, I do not believe that the vector-/component-wise clipping (as in the paper) will provide robustness in the training trajectory due to estimating the median gradients (yet there might be some other reasons).
>
> We agree that the interleaving of updates makes this a delicate matter. However, we think that we made several attempts in the numerical section of the paper to answer this question: we compare VClip and CClip to their full median counterparts, when then weights are updated as well (Figure 4). One can see that VClip/CClip perform in the same order as l1/l2 median, in particular the order flips from (S1)/(S2) to (S3). This, together with the theoretical background, strongly suggests that the performance of VClip/CClip relates to the properties of the median that they track.
>
>
> > The paper lacks such an explanation that the propositions are generally true for any kind of stochastic gradient method with biased gradient estimators.
>
> Thanks for pointing this out. Indeed, we only mention this in the Appendix, see Appendix A.2, first sentence:
>
> >  The following proposition is an application of biased SGD results to the SMGD algorithm with error.
>
> We will make this clear in the main text in the revised version.
>
> Re Weakness 2, subpoint 2: in general the answer to this depends on the exact setting and assumptions. But we believe that the questions posed here are already answered in previous works, which we refer to in our paper (see next response).
>
> > Do we have a theoretical/rigorous guarantee that the methods using the sample mean gradients always fail to converge under heavy-tailed (or state-dependent) gradient noise?
>
> Yes, see the result in Zhang et al., 2020 b, Remark 1. We refer to this result explicitly in the revised paper.
>
> > Exactly when (e.g., for what class of distributions, or with how large a sample size) does the variance of the estimator become reasonably small in general?
>
> See the discussion below Proposition 5.4, in particular we point to footnote 2: for example for the Cauchy distribution the variance of the sample median is finite under mild conditions, whereas for the mean it is always infinite. Of course, the answer to this depends on the exact distribution and other assumptions.
>
> > Is it always true that SGD using the sample median gradients succeeds in converging when SGD using the sample mean gradients can succeed?
>
> We are not sure if we understood this question correctly. We interpret it as: is there a general theory that shows that if the sample mean gradient achieves X, then so does the sample median? In this case, we do not have an answer beyond the implications of the biased SGD analysis applied to sample mean and median.
>
> >  I think we need convergence metrics other than “mean” squared gradient norm or “mean” sub-optimality in function values
>
> We think alike, but we believe that devising a median-aware convergence proof for SGD is a separate challenge which is beyond the topic of this paper.
>
> > By the way, the step size conditions in Propositions 5.3 and 5.4 are not reflected in Corollary 5.6.
>
> Thanks for catching this, we will update accordingly.
>
> > the descriptions of Propositions 5.3 and 5.4 lack clear dependencies regarding the sample size
>
> The dependence on sample size is reflected in the constants $\sigma_1$ and $\sigma_2$. However, as the dependence of these parameters on sample size is in general different for each distribution, and we did not want to restrict the applicability to a single distribution, we opted for the current presentation. Please note that Corollary 5.6 translates the general theorem to a concrete family of distribution, and in that case we do have an explicit dependence on sample size.
>
> > a more interesting topic would be the behavior of optimization algorithms when applying these clipping techniques
>
> Could you explain this comment in more detail? We already performed several numerical comparisons of the optimization algorithms with all these different clipping techniques in Section 6.2 (least squares and language model training).

---

> > ### Author Response · Authors · 2025-09-18
> > **Author Response Part II**
> >
> > >  Is the median estimation really happening when we interweave the clipping-based SPP update and the weight update?
> >
> > Once we start to interweave updates, we have no theoretical guarantee for convergence to the median. For the methods presented in Section 4, we can use the estimation to median connection as a principle of algorithm design, even though we do not have theoretical guarantees. We mention this explicitly in the last paragraph of the section (“Outline of remaining paper”). Further, our experiments demonstrate that indeed the online median/clipping methods perform in strong correlation to their respective sample median estimator (see Figure 4).
> >
> > When revising this point, we realized that the claim of *"hidden estimation of the median"* in the introduction might be misleading, hence we rephrased this.
> >
> > > How good/bad/similar are they compared to SGD using the sample mean/median gradients?
> >
> > We believe this question is answered by Figure 4. However, we might have misunderstood the question, in which case we would kindly ask you to clarify.
> >
> > > The only thing I can expect from Propositions 5.3 and 5.4 is that V/CClip-SGD are strictly worse than SGD with exact sample median gradients.
> >
> > This is only true if the usual variance conditions necessary for (mean) SGD are satisfied. But this is exactly the point of our theory: there are scenarios where the variance assumptions for SGD are not met, but indeed they can be satisfied for the median (see Corollary 5.6 for example). We do agree that in scenarios where the assumptions of SGD are easily met (light tails), we can not expect an advantage of using the median in general.
> >
> > > What happens for $\|\cdot\|^\gamma$?
> >
> > This is an interesting suggestion. In terms of computation, the proximal operator of such functions raises some difficulties, as no closed forms are known in general, and this would require running a subroutine. It is reasonable to expect that between 1 and 2 the estimator will interpolate between median (gamma=1) and mean (gamma=2). For gamma< 1, the distance function would be non-convex, and hence require additional techniques. Due to these additional challenges, we did not explore this direction in further detail.
> >
> > > Why is momentum worse for $\alpha=2$?
> >
> > This effect is due to using a step size that is not tuned individually for each method. We add a plot with smaller $\tau$ which shows that for $\alpha=2$ momentum can be better (see Figure 8 in revision). However, the main takeaway from this experiment is that momentum quickly becomes unstable when $\alpha$ decreases (=tails become heavier); this effect is observed equally for the smaller value of $\tau$.
> >
> >
> > > What will happen in the plots if we encounter a skewed (but heavy-tailed) noise distribution, having the median different from the mean?
> >
> > Thank you for the suggestion, we added this ablation in Figure 8. As expected, when the noise is skewed the V/CClip methods also cannot properly denoise the samples, as the noise median is not zero anymore.
> >
> > > Manipulating the value of the parameter $\mu$ for the Huber function?
> >
> > We added a study in Figure 9 in the appendix. Overall, the impact of mu on the performance seems rather small, and the value of 1.345 proposed originally by Huber seems to be a good rule of thumb.
> >
> > > Will a smaller learning rate or a learning rate scheduling alleviate the instability in SGD with sample mean gradients?
> >
> > No, SGD still fails to converge for smaller learning rates when the noise is heavy-tailed (and/or state-dependent). This is in line with the theory in Zhang et al., 2020b.
> >
> > > Probably, l1-SMGD can also be tested for training language models.
> >
> > Yes, this would be feasible but not straightforward as it would require changing the standard Pytorch aggregation over the batch (from mean to median). Due to this technical challenge, and the rather disappointing performance of componentwise clipping (the online version of l1 median)  we saw for the language tasks, we did not follow this direction further.
> >
> > > Adam with a usual gradient clipping, VClip, or CClip can be put into the baselines.
> >
> > Our main goal of adding Adam here was to show that only changing the clipping technique does not fully close the gap of SGD-type methods to Adam. It is not straightforwardly clear to us how to combine Adam-type preconditioning with clipping *in a principled way*, and we think this is interesting future work.

---

### Review · Reviewer_pgqA · 2025-09-09

**Summary Of Contributions:**

This submission concerns clipping and robustness in optimization. Clipping gradients is commonly used in practice to make an optimization process less sensitive to noise. The main contribution of this paper is to show how clipping can be viewed as a robustified momentum procedure. The submission has some derivations (showing the equivalences), convergence statements, and experiments.

In more detail, we can view standard momentum as computing an exponentially weighted moving average of the gradients. As I understand it, this paper asks: what if we used some other notion of "average gradient" as our momentum term? They suggest using online, robust methods for a median (based on stochastic proximal point solving the associated M-estimation task). They then show how existing methods such as clipping can be viewed as instantiations of this approach. This provides some formal explanation for their success in many practical applications.

They provide theory showing how some of these approaches will converge in settings where the gradient noise is heavy-tailed. There are experiments on synthetic data and a language modeling task.

**Audience:**

Yes

**Claims And Evidence:**

Yes

**Requested Changes:**

- Modify the warmup in Section 3.1 to make it clear which parts of the presentation are new.
- If you believe it is appropriate, please consider adding more context around the convergence results.

**Strengths And Weaknesses:**

The submission has a very interesting narrative and is carefully written. It tackles a fundamental topic. Surely many readers will find it interesting.

I view this work as less focused on new methods and more about revealing a new perspective. This type of work could be very impactful. However, I am not an expert in this area and it is hard for me to evaluate this.

At times, I felt unsure of which results were new and how they compare to what is known. For example, in Section 3.1 it says "we will derive a new perspective on SGD-M, namely being an *online estimator of the mean gradient*." (Emphasis theirs.) Now, this section is explicitly a warmup, but the perspective is not new: one can see from the description that $m_t$ is an exponentially weighted moving average of the past gradients. See [1,2] for explicit discussion of this. I believe the new connection is running the framework through SPP, but this could be more explicit.

Similarly, in Section 5 I was left wondering what the existing similar results are. I see that the methods converge under reasonable heavy-tailed noise assumptions. Do these results match rates from an easier setting? Are they the best results one could hope for? Do they give new guarantees for an existing algorithm?

[1] https://optimization.cbe.cornell.edu/index.php?title=Momentum

[2] https://stats.stackexchange.com/questions/353833/why-is-gradient-descent-with-momentum-considered-an-exponentially-weighted-avera

---

> ### Author Response · Authors · 2025-09-18
> **Author Response**
>
> We thank the reviewer for the positive feedback and encouraging review of our paper. We implemented the requested changes from this and the other reviews (highlighted in red in the updated PDF).
>
> > I believe the new connection is running the framework through SPP, but this could be more explicit.
>
> Correct, the new insight is that momentum is equal to running SPP for the mean estimation problem at fixed weights. This is different from seeing momentum as an exponential weighted average of past gradients. We clarify this in the respective section.
>
> > More context for the results in Section 5
>
> In terms of proof technique, our result is similar to Theorem 3 from Demidovich et al. (2023) (we do obtain better constants though). This was explained in Appendix A.2 and will now be explained in the main body.
>
> In terms of noise assumptions, the main distinction to previous results is that we show that SGD with the sample median still works under state-dependent noise (see the discussion below Assumption 5.5 for details).
>
> Regarding the question whether these are the best results one can hope for, this is in general hard to answer as it would require lower bounds for a specific set of function class, algorithm, and noise assumptions. We therefore believe that this question is beyond the scope of this paper.

---

> > ### Comment · Reviewer_pgqA · 2025-09-23
> >
> > Thank you for the response. After reading other reviews and responses, I maintain my original (largely positive) view of the submission.
> >
> > One extremely minor point: the new red text in Section 3.1 says
> > >It is easy to see that momentum performs an exponentially weighted average over past gradients. In this section, we will derive a new perspective on SGD-M, namely being an online estimator of the mean gradient at a given point.
> >
> > I still find this phrasing strange: I would already call exponential weighting an online estimator of the mean? (I think I understand why you draw the distinction.)

---

> > > ### Author Response · Authors · 2025-09-24
> > > **Author Response**
> > >
> > > Dear reviewer,
> > >
> > > we are happy to hear that all concerns have been resolved.
> > >
> > > Regarding the sentence in Section 3.1: indeed the new perspective is the connection to stochastic proximal point. We will try to clarify this even more in the final version. In any case, this section serves only as warmup to derive the following parts, hence it is a minor point.

---

### Author Response · Authors · 2025-09-18
**General Response to Reviewers**

Dear Action Editor and Reviewers,

thank you for taking the time to review our submission. We have now responded to all questions and points raised in the reviews. Please let us know in case we missed something, or misunderstood a question.

We also uploaded a revised PDF, all **changes are highlighted in red**. We added some additional experiments and ablations in Appendix C.1.

We are looking forward to further discussion.

---

### Decision · Action_Editor_UPMS · 2025-10-07

**Recommendation:** Accept with minor revision

**Additional Comments:**

For the most part, I think the paper is acceptable as-is. All of the reviewers agreed that the paper was interesting and well-written, and I agree with them. Of the three reviewers, the final recommendations were "Accept", "Leaning Accept", and "Leaning Reject". For reference, the final recommendation comments from the reviewer who voted "Leaning Reject" were as follows:

> Although I find this paper very interesting, I am still not fully convinced about the main claim of the paper, even after the author's rebuttal. I believe the main claim was "estimating the median gradient over training trajectory using some kind of clipping iterate-gradient gap provides more robustness in the training trajectory." However, this claim is neither rigorously/fully proved in mathematical theory nor verified in comprehensive numerical experiments.

> In my opinion, the main theoretical contributions are restricted to (1) showing the relationship between some form of clipping and the median estimation of random vectors and (2) showing a (quite imperfect) convergence guarantee (based on non-median-based convergence criteria) of biased SGD with median gradient estimates. The actual theoretical analyses of the optimization algorithms based on VClip and CClip are not presented. At least, I want to recognize the significance of the theoretical analysis part only when the median-based convergence proof is possible in this paper. Otherwise, I do not think the theory in it tells much about its main claim.

> On the other hand, the authors claim that they have numerically showcased some "relationships" (yes, "correlation", not "causation") between the training robustness and clipping-based median gradient estimation. Nonetheless, in my view, the scope of numerical verification is quite restrictive. Considering some limitations in the theoretical analysis (as the authors have admitted), I hope they could include more abundant examples of numerical validations.

I understand where the reviewer is coming from with these comments, and the comments are not untrue in my opinion. That said, I disagree that *"estimating the median gradient over training trajectory using some kind of clipping iterate-gradient gap provides more robustness in the training trajectory"* is the actual main claim made by the authors. I think this is an idealized main claim that would be great to have in future work, but I believe the summary of contributions in the present manuscript, plus the detailed comments interspersed throughout the text indicate that the authors are aware of the limitations of their work and open about them. As such, the main claims as I understand them are solid, and thus coupled with the fact that all reviewers found the paper appealing and clear, I think it is safe to recommend this paper for acceptance.

Dear authors: I have recommended "acceptance with minor revision"; in the summary of contributions, the sentence *"Our experiments also show that our online median estimates are robust, while being less expensive to compute."* Less expensive than what? What is being compared here, and how is it measured? Please clarify and re-write the relevant paragraph.

**Audience:**

Yes

**Audience Explanation:**

Stochastic gradient-based methods are the workhorse of modern machine learning, and situations with potentially small and noisy samples mean that robustness to outliers is of great practical importance, in addition to the theoretical points of interest that arise when we consider a heavy-tailed feedback scenario. The paper is well-written and has a clear narrative; I think the paper will have an audience in TMLR.

**Claims And Evidence:**

Yes

**Claims Explanation:**

In the context of gradient-based learning algorithms under potentially heavy-tailed stochastic gradients, the authors use a stochastic proximal point (SPP) formulation to show how certain kinds of gradient clipping can be linked up to a sub-routine which "tracks" the median gradient over iterations. While there is some dispersion in the opinions of the reviewers regarding what the "main claims" of this paper are, my opinion is that their main claims are centered around two main points: (a) that they have elucidated connections between gradient clipping and median tracking (via SPP), and (b) they show that the median trackers allow for convergence guarantees under heavy tails (assuming the sample grad median can be computed accurately enough). With these two points as their main claims, I feel the evidence presented is sufficiently clear and accurate. The exposition is well-written, the paper is easy to follow, and the formal results are presented well.